# Learning View-invariant World Models for Visual Robotic Manipulation

**Jing-Cheng Pang**[1,2,3,*], **Nan Tang**[1,3,*], **Kaiyuan Li**[1,3], **Yuting Tang**[4,2], **Xin-Qiang Cai**[2],
**Zhen-Yu Zhang**[2], **Gang Niu**[2], **Masashi Sugiyama**[2,4] **& Yang Yu**[1,3,◇]

[1] National Key Laboratory for Novel Software Technology, Nanjing University, China &
School of Artificial Intelligence, Nanjing University, China;
[2] RIKEN Center for Advanced Intelligence Project, Japan;
[3] Polixir.ai, China; [4] The University of Tokyo, Japan

## ABSTRACT

Robotic manipulation tasks often rely on visual inputs from cameras to perceive the environment. However, previous approaches still suffer from performance degradation when the camera's viewpoint changes during manipulation. In this paper, we propose ReViWo (Representation learning for View-invariant World model), leveraging multi-view data to learn robust representations for control under viewpoint disturbance. ReViWo utilizes an autoencoder framework to reconstruct target images by an architecture that combines view-invariant representation (VIR) and view-dependent representation. To train ReViWo, we collect multi-view data in simulators with known view labels. Meanwhile, ReViWo is simultaneously trained on Open X-Embodiment datasets without view labels. The VIR is then used to train a world model on pre-collected manipulation data and a policy through interaction with the world model. We evaluate the effectiveness of ReViWo in various viewpoint disturbance scenarios, including control under novel camera positions and frequent camera shaking, using the Meta-world & PandaGym environments. Besides, we also conduct experiments on real world ALOHA robot. The results demonstrate that ReViWo maintains robust performance under viewpoint disturbance, while baseline methods suffer from significant performance degradation. Furthermore, we show that the VIR captures task-relevant state information and remains stable for observations from novel viewpoints, validating the efficacy of the ReViWo approach.

## 1 INTRODUCTION

Developing robots capable of completing various manipulation tasks is a hallmark of machine intelligence. Previous approaches in this domain often rely on visual inputs from cameras to perceive the environment, with a policy trained on these visual inputs using reinforcement learning (RL) algorithms and pre-collected robotic manipulation data (Kalashnikov et al., 2018b; Quillen et al., 2018; Agarwal et al., 2020; Levine et al., 2020; Pang et al., 2024). However, the policy learned in this way often suffers from performance degradation when merely changing the camera position during deployment (Liu et al., 2024). This challenge, referred to as the *viewpoint disturbance*, arises from the policy's learned representations, which fail to separate the view-invariant information from the observation: they encode the entirety of the observations, including both the view-invariant task state and the view-dependent information like light. As a result, the changes in the viewpoint can lead to a substantial transformation in the learned representations, which can compromise the policy's effectiveness when exposed to novel viewpoints (Liu et al., 2024).

There are existing works focusing on training robust robotic manipulation policies under viewpoint disturbances. For example, MVWM (Seo et al., 2023) tries to train a robust image representation using a multi-view masked autoencoder, by encoding a masked image and reconstructing target images in different viewpoints. The learned representation is then utilized for policy decision-making.

---

*Equal contribution. Full authorship contribution statements appear after the main text. ◇ Corresponding: yuy@nju.edu.cn.

However, the learned representation can still be sensitive to variations in viewpoint because it relies on a single encoder model to extract information, and then reconstruct complex, viewpoint-dependent visual information which can be highly challenging. RT-X series works (Brohan et al., 2023; Zitkovich et al., 2023; O'Neill et al., 2024) attempt to address viewpoint disturbance by training policies on extensive data. RoboUniView (Liu et al., 2024) learns a unified representation from 3D multi-view images using an autoencoder (Bank et al., 2023), but it requires all viewpoints to contain similar information, and relies on 3D data with camera calibration. Although these approaches have demonstrated a degree of effectiveness in addressing viewpoint disturbances, they have yet to achieve the complete and effective decoupling of view-invariant state information from the viewpoint, which is important for robust robotic control.

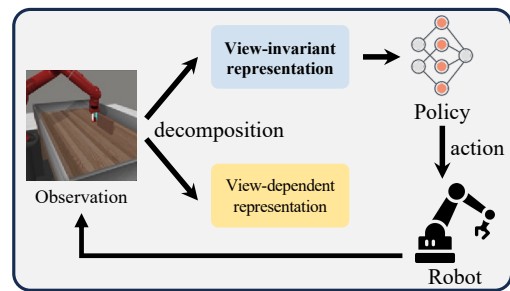

In this paper, we investigate robust robotic manipulation under viewpoint disturbance. Motivated by human's ability to adapt to changes in viewpoint, a skill that is supported by cognitive science research in human psychology (Souto & Kerzel, 2021), we propose to decompose the visual observation into a *view-invariant representation* (VIR) for robot control, as shown in Fig. 1. More specifically, we propose Representation learning for the View-invariant World model (ReViWo), which employs two encoders to separately extract view-invariant and view-dependent

Figure 1: An illustration of the robotic manipulation with view-invariant representation.

representations from the image, and a decoder to reconstruct a target image that combines the view-invariant and viewpoint information from two encoders' inputs, respectively. This training process requires the *view labels*, which can be obtained when collecting multi-view data from fixed-position cameras. Leveraging the learned VIR, ReViWo trains a world model to learn the environment dynamics from the offline manipulation data. Finally, ReViWo trains policy with RL algorithm and using the world model.

Our contributions are as follows: First, we introduce an *observation decomposition* idea for enhancing visual robotic manipulation under viewpoint disturbances, which enables the robot to focus on invariant information across viewpoints. This decomposition distinguishes this work from previous works that process the observation into a single representation. Second, we implement this idea by proposing ReViWo, comprising a world model and an autoencoder framework that learns the *view-invariant representation* from pre-collected multi-view data with view labels. Besides, we also train ReViWo with *Open X-Embodiment* (O'Neill et al., 2024), to incorporate the knowledge from more diverse robotic manipulation data. Lastly, we conduct extensive experiments to verify the efficacy of the proposed method: Meta-world (Yu et al., 2019) and PandaGym (Gallouédec et al., 2021) simulation environments and real world ALOHA robotics arm. The experiment results demonstrate the robustness of ReViWo in the face of two types of viewpoint disturbance: a novel camera installation that results in $10 \rightarrow 90$ degrees of viewpoint offset from the training setting, and a scenario with continuous camera shaking.

## 2 BACKGROUND

**RL and Offline RL.** We consider an RL problem aiming at learning a policy that maximizes the expected cumulative discounted reward in a Markov Decision Process (MDP) (Puterman, 1994; Sutton & Barto, 1998; Yuan et al., 2023; Cai et al., 2023b), which is represented by the tuple $\mathcal{M} = (\mathcal{S}, \mathcal{A}, \mathcal{P}, \mathcal{R}, \gamma)$. In this tuple, $\mathcal{S}$ denotes the state space, $\mathcal{A}$ the action space, $\mathcal{P} : \mathcal{S} \times \mathcal{A} \rightarrow \mathcal{S}$ the transition function of the environment, $\mathcal{R} : S \times \mathcal{A} \rightarrow \mathbb{R}$ the reward function that evaluates the quality of the agent's action, and $\gamma \in (0, 1)$ the discount factor which balances the immediate and future rewards. A policy $\pi : \mathcal{S} \rightarrow \mathcal{A}$ defines the agent's strategy, mapping states to a distribution over possible actions. The RL agent interacts with the environment as follows: at each timestep $t$, the agent observes a state $s_t$ from the environment. It then selects an action $a_t$ based on the policy $\pi(\cdot|s_t)$ and executes it in the environment. Next, the agent receives a reward $r_t$ and the environment transitions to a new state $s_{t+1}$ according to the transition function $\mathcal{P}(\cdot|s_t, a_t)$. The objective of RL is to find a policy that maximizes the expected sum of rewards over time: $\mathbb{E}_\pi[\sum_{t\geq0} \gamma^t r_t]$. In offline RL, we only have access to pre-collected interaction data: $\{(s_t^i, a_t^i, r_t^i, s_{t+1}^i)_i\}$ and no further

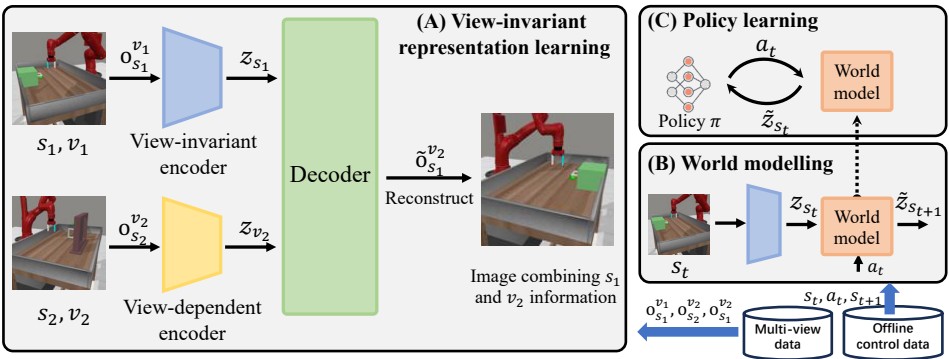

Figure 2: Overall training framework of ReViWo method. **(A)** View-invariant representation learning process trains two encoders to decompose view-invariant and view-dependent information within the images, and a decoder to reconstruct the target image. **(B)** The World modeling process builds a world model upon the learned VIR. **(C)** Policy learning by interacting with the world model.

environment interacting is allowed. The objective of offline RL is to find a policy $\pi(a_t|s_t)$ that, when deployed, maximizes the cumulative reward within the confines of the MDP as characterized by the dataset.

**VAE and VQ-VAE.** Variational Autoencoder (VAE) (Kingma & Welling, 2014) is an effective tool to extract information from the image, which learns to encode an image into a continuous latent space and subsequently reconstruct the image from this space. VAE represents an image with a latent vector that captures the essential information required to reconstruct the image. We use $o \in \mathbb{R}^{H \times W \times C}$ to denote the image, where $H$, $W$, and $C$ represent the height, the width, and the number of channels of image $o$. The encoder network $q_\phi(z|o)$ maps the input image to a distribution over the latent space, and the latent vector $z$ is sampled from this probability density: $z \sim q_\phi(z|o)$. The decoder network $p_\theta(o|z)$ aims to reconstruct an image from the latent vector. The objective of a VAE is to minimize the negative log-likelihood as below:

$$\mathcal{L}_{\text{VAE}}(\theta, \phi) = \mathbb{E}_{q_\phi(z|o)}[-\log p_\theta(o|z)] + D_{KL}(q_\phi(z|o)||p(z)), \tag{1}$$

where the Kullback-Leibler divergence term regularizes the latent space by enforcing similarity to a prior distribution $p(z)$, typically chosen to be a standard normal distribution.

Vector Quantized VAE (VQ-VAE) (van den Oord et al., 2017) extends VAE by introducing a discrete latent space, which can be more suitable for information extraction. VQ-VAE introduces two main modifications: (a) The continuous latent vector is discretized by mapping it to the nearest vector in a learnable codebook $e = \{e_1, e_2, ..., e_K\}$. This mapping results in a discrete latent variable $z_q$, which is used for reconstruction; (b) The training objective includes a codebook loss. The codebook loss consists of two terms: a commitment loss that encourages the encoder outputs to commit to a codebook vector, and a quantization loss that moves the codebook vectors towards the encoder outputs. The overall loss function is:

$$\mathcal{L}_{\text{VQ-VAE}}(\theta, \phi; o) = \mathbb{E}_{q_\phi(z|o)}[-\log p_\theta(o|z)] + \|\text{sg}[q_\phi(z|o)] - e\|_2^2 + \|q_\phi(z|o) - \text{sg}[e]\|_2^2, \tag{2}$$

where $\text{sg}[\cdot]$ denotes the stop-gradient operator.

## 3 METHOD

This section presents the proposed ReViWo method. The main idea of ReViWo is to decompose the visual observation into view-invariant representation (VIR) and view-dependent representation (VDR) separately. In the next, we give a formal problem definition and then elaborate on the two key steps of ReViWo method: (1) *view-invariant representation learning* and (2) *world modeling and behavior learning*.

### 3.1 PROBLEM FORMULATION

This study focuses on training a policy for visual robotic control, which is robust to viewpoint disturbances. Consider we have the following two datasets:

1. *A multi-view dataset $\mathcal{O} = \{(o_{s_i}^{v_1}, o_{s_i}^{v_2}, \ldots, o_{s_i}^{v_N})_i\}$, where $v_i \in \mathcal{V}$ is a specific viewpoint within the set of viewpoints $\mathcal{V}$, $o$ denotes a visual observation, $N$ the number of viewpoints, and $s_i$ the the state presented in the image. Here we have multiple trajectories from $N$ fixed viewpoints. In practical, we can collect this dataset by recording robotic manipulation with multiple cameras.*

2. *A robotic manipulation dataset $\mathcal{D} = \{(s_t^i, a_t^i, r_t^i, s_{t+1}^i)_i\}$, where the state $s_t^i$ is visual observation from a fixed position camera, $a_t^i$ is robot behaviors and $r_t^i$ denotes the observed reward.*

The primary objective is to derive a policy that maximizes the rewards on visual observations of novel viewpoints that are not present in the manipulation dataset, thus ensuring the policy's decision-making capability is not limited to the viewpoints within the training data. We will provide a detailed discussion in Sec. 4.1 on how to construct *viewpoint disturbance* in the experiments.

### 3.2 VIEW-INVARIANT REPRESENTATION LEARNING

The first step of ReViWo is to learn VIR using multi-view images. The main idea is to separate the view-invariant and view-dependent information within the image and find a training objective to enable both two representations to be meaningful. Then the policy can make decisions based on the view-invariant information, which is robust to the viewpoint changes. ReViWo learns the representation using an autoencoder architecture, as shown in Fig. 2 (A). The framework comprises two encoders and a decoder, where each encoder accounts for capturing specific information within an image, and the decoder reconstructs the images from the encoders' output.

**Encoders for information extraction.** ReViWo utilizes two encoders to extract view-invariant and view-dependent from visual images separately. Specifically, we use $q_\phi^S(o)$ to represent the view-invariant encoder (VIE) and $q_\phi^V(o)$ for view-dependent encoder (VDE), where $\phi$ denotes the parameters of the autoencoder model. The encoders are implemented upon the vision transformer (ViT) architecture (Dosovitskiy et al., 2021), which processes an image by partitioning it into a sequence of patches. Each patch is converted into a token embedding via a learnable convolutional neural network (CNN). These token embeddings are then concatenated with their corresponding positional embeddings, which are computed using the function $(1 - \texttt{timestep})$, to form the inputs for the transformer at each timestep. We present our implementation of ViT architecture in Fig. 9. For the VIE, the transformation can be formalized as $z_s = q_\phi^S(o_s^v)$, where $o_s^v \in \mathbb{R}^{M \times H \times W \times 3}$ denotes the input image patches and $z_s \in \mathbb{R}^{M \times C}$ represents the outputted VIR. Here, the dimensions $W \times H$ denote the size of each image patch, $M$ the patch number and $C$ denotes the dimensions of the output feature space. Thus the output can be regarded as a sequence of features corresponding to each input patch. The VDE employs the same architecture as the VIE and yields an output $z_v = q_\phi^V(o_s^v)$, capturing the view-dependent features.

**Decoder for image reconstruction.** The decoder is also a transformer-based architecture, which inputs both view-invariant and view-dependent features and outputs a reconstructed image: $\tilde{o}_s^v = p_\phi(z_s; z_v)$. The features $z_s$ and $z_v$ could be extracted from different images within the same domain, and the goal for the decoder is to generate a novel image that integrates the features of the two inputs. The decoder's input is the concatenated outputs from the two encoders. Instead of generating the image patch by patch in an auto-regressive manner, we follow the Genie approach (Bruce et al., 2024) to output all patches simultaneously to enhance efficiency. Each output feature is then processed by a CNN to form an image patch, with the collective patches constituting the entire image.

**Training objective.** The autoencoder is trained to reconstruct pixels following a VAE-like objective (Kingma & Welling, 2014), which involves optimizing the parameters $\phi$ by minimizing the negative log-likelihood as in Eq. (3):

$$\mathcal{L}_{\text{AE}}(\phi) = \mathbb{E}_{o_{s_i}^{v_j}, o_{s_m}^{v_n}, o_{s_i}^{v_n} \sim \mathcal{O}} \left[ -\log p_\phi(o_{s_i}^{v_n} | q_\phi^S(o_{s_i}^{v_j}); q_\phi^V(o_{s_m}^{v_n})] + \lambda_1 \mathcal{L}_{\text{VQ}}(\phi) + \lambda_2 \mathcal{L}_{\text{Contrastive}}(\phi). \quad (3)$$

The first term, the image reconstruction loss, measures the model's ability to predict an image of a given state from a novel viewpoint, by using state information from one image and viewpoint information from a different image. Minimizing this term enables the model to disentangle state and view information, as it must accurately predict the target image by fusing these two distinct information types. The reconstruction of the image, $o_{s_i}^{v_n}$ conditioned on the state-related features from $o_{s_i}^{v_j}$, trains

the model to concentrate on certain aspects in the image that are invariant to viewpoint changes, such as the state of the object. The inclusion of $z_v$ in the reconstruction encourages the model to learn viewpoint-specific features, such as lighting, angle, background, and perspective, which are independent of the object's state. The second term, $\mathcal{L}_{\mathrm{VQ}}$, is from VQ-VAE (van den Oord et al., 2017), which is known for its efficacy in image reconstruction. It comprises a commitment loss that ensures the encoder's output vectors are proximate to their nearest vectors in the quantization dictionary and a quantization loss that ensures the quantized vectors accurately represent the encoder's outputs. We refer readers to Eq. (2) for VQ-VAE loss. The term $\mathcal{L}_{\mathrm{Contrastive}}$ is designed to optimize the variance of the latent representations $z_s$ and $z_v$ for multi-view images under a variety of conditions. This term encourages $z_s$ remain consistent across identical states and varies across different states. Similarly, it ensures that $z_v$ is consistent for identical viewpoints and varies for different viewpoints, thereby enhancing the model's ability to capture the nuances of multi-view image representations. Refer to Appendix A for the mathematical justification that Eq. (3) can effectively separate view-invariant and view-independent representations.

**Integration of Open X-Embodiment data without view labels.** In addition to the data with view labels, we also involve multi-view data without view labels from the Open X-Embodiment dataset (O'Neill et al., 2024), which are readily available on the internet. This dataset encompasses a wide array of robotic manipulation tasks performed by various robots, thereby enriching the training of the autoencoder with a broader spectrum of information and diversity. Note that in the absence of labels, the inputs to the VIE and the VDE correspond to identical states observed from different viewpoints. To prevent the VDE from encoding the entirety of the image information for image reconstruction, we introduce a weighting factor in the loss calculation for these unlabeled data.

### 3.3 World Modelling and Policy Learning from Offline Manipulation Data

**View-invariant representation as the state for visual control.** The previous section introduces how ReViWo learns the VIRs, which capture the invariant features of the environment that are essential for decision-making and are not affected by viewpoint changes. This is similar to how humans focus on the relevant aspects of a task, regardless of how they are viewing the scene. By focusing on these invariant features, the policy can make decisions based on the true state of the environment rather than on extraneous visual details that may change with the camera angle. Therefore, we naturally regard the VIR as the task state for control. Building on this foundation, ReViWo proceeds to train a world model and policy via offline reinforcement learning.

**World modeling and behavior learning.** We employ the COMBO algorithm (Yu et al., 2021) for world modeling and behavior learning, which is an offline model-based reinforcement learning methodology. COMBO algorithm first trains an ensemble of world models, utilizing ensemble methods (Ganaie et al., 2022) to improve the prediction accuracy and robustness of the world model. Then, the policy interacts with the world model, from an initial state that is sampled from the offline dataset, for a rollout. The policy is optimized by utilizing a combination of the real offline dataset and the simulated rollouts generated by the world model. The world model $\mathcal{M}_\theta(q_\phi^S(s_t), a_t)$, parameterized by $\theta$, is trained by minimizing the difference between the predicted next state with the target state, as shown in Eq. (4):

$$\mathcal{L}_{\mathrm{WM}}(\theta) = \mathbb{E}_{(s_t, a_t, s_{t+1}) \sim \mathcal{D}} [-\log \mathcal{M}_\theta(q_\phi^S(s_{t+1}) | q_\phi^S(s_t), a_t)]. \tag{4}$$

In addition to the world model, we also train a reward model by supervised learning on the offline control data, which is capable of predicting the rewards for given state-action pairs. This reward model provides the necessary reward signals for the policy's learning process.

## 4 Experiment

In this section, we conduct extensive experiments to evaluate our proposed method for robust control under viewpoint disturbance. The goal of the experiments is to answer the following key questions: (1) How does ReViWo perform compared to existing baseline methods under viewpoint disturbance (Sec. 4.2)? (2) How is the algorithm's robustness on viewpoints variations, and how much data is required to train ReViWo(Sec. 4.2)? (3) Does ReViWo learn meaningful view-invariant and view-dependent representation (Sec. 4.3)? (4) What is the impact of each component in ReViWo on the overall performance of the algorithm (Sec. 4.4)? We first introduce the experiment setup.

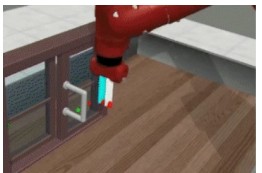 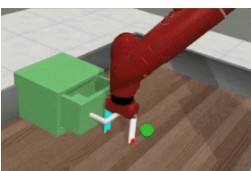 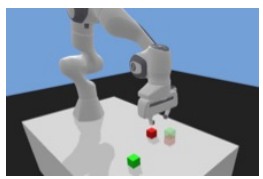 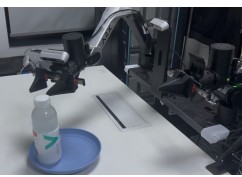

       (A) Meta-world          (B) PandaGym       (C) ALOHA

Figure 3: A visualization of the environments in our experiments. **(A)** In the Meta-world, the agent controls a Sawyer robot to manipulate various objects such as door, drawer and window. **(B)** In the PandaGym environment, the agent controls a Franka robot with 7-DoF. **(C)** ALOHA robot that manipulates a bottle to the plate.

## 4.1 EXPERIMENT SETUP

**Evaluation environments.** We conduct experiments on two robotics manipulation environments: Meta-world (Yu et al., 2019) and PandaGym (Gallouédec et al., 2021), as shown in Fig. 3. **(1) Meta-World:** This environment requires the agent to control a Sawyer robotics arm with 7 degrees of freedom (DoF) and a parallel finger gripper. The action space is a 2-tuple. The first element of the tuple is the displacement in 3D space of the end-effector, and the second is a normalized torque to be applied by the gripper fingers. Meta-World offers a suite of 50 distinct manipulation tasks, covering a wide array of scenarios, such as interactions with doors, windows, drawers, and balls. For our experiments, we assess the performance of our methods on a subset of tasks: door opening, drawer opening, and window closing. **(2) PandaGym:** This environment involves a Panda robotic arm by Franka Emika, which has 7 DoF and a parallel finger gripper. PandaGym is primarily focused on block manipulation tasks, which are designed to test the robot's foundational skills, such as reaching a goal point. The action space is defined by the gripper's movement command, which includes three coordinates for spatial movement, and the finger movement, which is a single coordinate reflecting the gripper's aperture. For certain tasks, the gripper is fixed in a closed position, which reduces the action space to only the gripper's spatial movement command. **(3) ALOHA**: We include a real world ALOHA robot to manipulate a bottle to a plate. The task involves three stages: reaching for a bottle (stage 1), grasping it (stage 2), and then placing it on a plate (stage 3).

We mainly use Door Open, Drawer Open, and Window Close from Meta-world, and Reach from PandaGym for evaluation, and we also conduct experiments on Coffee Button, Faucet Open, Dial Turn (Appendix F.1). The observations on all tasks are images with $128 \times 128$ pixels, which are captured by the fixed-position third-person camera. To introduce viewpoint disturbance, we can modify camera parameters, such as its azimuth, pitch, and height, to alter the viewpoint. For our experiments, we specifically alter the azimuth value, as this adjustment alone can introduce considerable visual perturbations and is sufficient to evaluate the robustness of the proposed methods.

**Dataset for training. Multi-view dataset** for training autoencoder includes three parts: (1) Meta-world data. We collect 51 trajectories of 17 tasks from Meta-world and record them utilize 20 cameras from different viewpoints, resulting in 1020 observation sequences (112040 observations) in total. The trajectories are sampled by pre-trained expert policies with Gaussian noise. (2) PandaGym. In this environment, we collect 30 trajectories of 5 tasks with 20 viewpoints, resulting in 600 observation sequences (30k observations) in total. (3) Open X-Embodiment (Routing Primitive Dataset (Luo et al., 2023)), including 4 viewpoints for manipulating multi-stage cable routing tasks. This dataset does not provide a viewpoint label. We utilize 101 trajectories in this dataset to keep a data balance, with 10064 observations in total. We refer readers to Appendix C.1 for the detailed setup of camera parameters for collecting data. For **offline control data**, we collect 400 trajectories on Meta-world, with a single viewpoint that doesn't exist in the multi-view data and 100 trajectories on PandaGym. Then, we train the world model and RL policy on two tasks separately. We investigate the algorithm's performance when collecting multi-view images on a narrower range of available viewpoints, and present the result in Sec. 4.4. For **real world** data collection, we collect a dataset of 128 trajectories using three cameras (two third-person cameras and one gripper camera). The dataset includes 10 different types of bottles and plates, which are placed randomly within the operational area during data collection. We use this real-world dataset to train both VIE and policy.

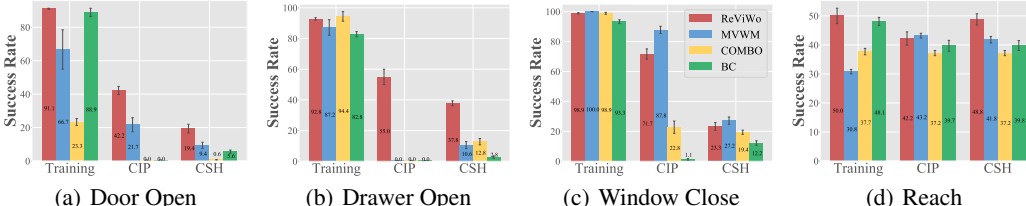

Figure 4: Performance of different methods under various viewpoint disturbances. The x-axis denotes the disturbance type, and the y-axis denotes the average success rate of the last two checkpoints, by evaluation for 30 episodes. The error bars stand for the half standard deviation over three random seeds. Results on more tasks are shown in Appendix F.1.

**Viewpoint disturbance setup.** In our experiments, we investigate the policy robustness under two distinct types of viewpoint disturbances: (1) Camera installation position (**CIP**): This category of viewpoint disturbance is designed to simulate the situation where the camera's installation position deviates from that of the policy training dataset. To this end, we adjust the camera's azimuth angle. Specifically, for the Meta-world experiments, we set the azimuth to a 10-degree offset, while for the PandaGym experiments, we apply a 90-degree offset. This alteration is intended to evaluate the robustness of policies to variations in the camera's installation angle. We also conduct experiments to investigate ReViWo's performance on various azimuth offsets (Fig. 5). (2) Camera shaking (**CSH**): This disturbance is relevant for scenarios with a hand-held camera, where the viewpoint can be continuously changing. To mimic this, we introduce a dynamic disturbance where the camera's azimuth angle is continuously adjusted during the robot's manipulation task. It is challenging because the model learns with a fixed viewpoint during training while being evaluated on a variable one. We present the visualization of the viewpoint disturbance in Appendix C.1.

**Implementation Details.** For autoencoder training, For world modelling and behavior learning for both ReViWo and baseline methods, we utilize OfflineRL-kit (Sun, 2023), a well-verified offline RL codebase. For all methods, the model is trained with an offline RL algorithm for 25000 gradient steps, and evaluated for 40 episodes. We conduct all experiments with four random seeds, and the shaded area or the error bars in the figures represent the standard deviation across four trials. We use 64 CPU cores (AMD EPYC 9654 @ 2.4GHz) and 4 GPUs (NVIDIA GeForce RTX 4090) for our experiments. More implementation details can be found in Appendix C.

## 4.2 PERFORMANCE UNDER VIEWPOINT DISTURBANCE

**Baselines for comparison.** We choose the following representative approaches in the domain of multi-view robotics manipulation or offline RL for comparison. (1) **MVWM** (Seo et al., 2023) handles the view disturbance by learning a multi-view masked autoencoder which reconstructs pixels of randomly masked viewpoints. It then learns a world model operating on the representations from the autoencoder. In MVWM's original implementation, the world model is learned in an online setting. We align it with our setting by training only with offline data and utilizing COMBO to train both world model and policy. (2) **COMBO** (Yu et al., 2021) is a model-based offline RL algorithm that trains a value function using both the offline dataset and the world model generated data while also additionally regularizing the value function on out-of-support state-action tuples generated via model rollouts. In the experiments, the COMBO method first pre-trains the vision representation on the same multi-view data as ReViWo using a standard VAE and then trains a policy that makes decisions based on the learned vision representation. (3) Behavior Cloning (**BC**) adopts a supervised learning approach to mimic the actions within the offline dataset. Similar to COMBO, we also use VAE to process the visual observation when implementing the BC baseline. All these baseline methods use the same multi-view dataset to pre-train vision representation with VAE. (4) Conservative Q-Learning (**CQL**) (Kumar et al., 2020) is an offline RL method. In our experiment in Sec. 4.4, we utilize CQL to train policy with the learned VIR, to verify the effect of the world model in ReViWo.

**Main results.** Fig. 4 presents the success rate of different methods under various levels of viewpoint disturbance. In general, our proposed ReViWo method outperforms the baseline methods across two types of viewpoint disturbances. While the baselines demonstrate considerable performance from the training viewpoint, they suffer from a clear decline when subjected to viewpoint variations.

This conclusion is supported by COMBO performance ($23.3 \to 0$ on Door Open, $94.4 \to 0$ on Drawer Open, and $98.9 \to 22.8$ on Window Close), which demonstrates the influence of viewpoint disturbance on these visual-based policies. BC shows promising performance on certain training viewpoints, yet its success rate substantially decreases under viewpoint disturbances. This performance degradation is due to the limitations of supervised learning on offline control data, which fails to preserve its robust performance when encountering novel observations. In contrast, ReViWo not only performs comparably to the training viewpoints and the CIP on tasks such as Door Open, but also demonstrates its robustness to viewpoint disturbances. This robustness is further evidenced by ReViWo's ability to handle tasks with a camera shaking, which is a significant challenge for policies to perform robustly (Seo et al., 2023). Interestingly, ReViWo achieves the highest performance on the training viewpoint for certain tasks. We hypothesize this is due to that VIR focuses on task-related domains within the observation and masks some unrelated information, thereby enhancing the policy learning for specific tasks.

**Impact of training data variations & Performance under various viewpoint offsets.** In this experiment, we study two important questions: the quantity of data necessary for training the autoencoder using ReViWo, and the degree of its robustness to changes in viewpoint. We train the autoencoder using images collected from Meta-world, employing two configurations: one with 10 random viewpoints across a 90-degree azimuth range (10V+90D), and another with 20 random viewpoints over a 180-degree azimuth range (20V+180D). For evaluation, we evaluate the algorithm's performance across various degrees of azimuth offset, ranging from 0 to 15 degrees. Fig. 5 shows the comparative performance of ReViWo trained with different data against various levels of camera viewpoint offset. Key findings include: (1) 10V+90D setting is enough for ReViWo to achieve a good performance, while additional data can further improve the algorithm's efficacy. For instance, the performance of ReViWo with 10V+90D is comparable to that with 20V+180D when the azimuth offset is $\leq 10$, and only marginally inferior when the offset $> 10$. In contrast, the COMBO method has a noticeable decline under even slight disturbance (e.g., $98.9 \to 55.6$ when the offset is 2.5); (2) ReViWo can effectively maintain the policy performance even when offset reaches 15 degrees, which is a considerable range during the practical robotic deployment. These insights serve as a guideline for applying ReViWo, and highlight its adaptability to viewpoint variations.

**Real world evaluation.** Tab. 1 presents the real world experiment results. We implement ReViWo-BC as follow: training VIE on the real world images and conducting behavior cloning on the control data with VIE representation. In this experiments, the policy inputs with all three camera images. To simulate viewpoint disturbances, we adjusted the azimuth by +15 degrees and the elevation by -15 degrees on two third-person cameras. We evaluate the method on both training and CIP viewpoint for 10 trajectories with random initialization. We also consider a baseline, ACT (Action Chunking Transformer) (Zhao et al., 2023), for comparison. ACT performs end-to-end imitation learning directly from real demonstrations, with pre-trained ResNet-50 as the vision encoder. The results indicate that the proposed method maintains robust control when subjected to novel viewpoints in a real-world setting, while imitation learning-based method suffers from a performance degradation Cai et al. (2023c;a). This result serves as a preliminary evidence of the applicability of ReViWo method. In contrast, ACT performs comparably to ReViWo-BC on training viewpoint, while exhibiting a significant decline in performance when viewpoint changes. These real world results further substantiate the efficacy of the ReViWo in handling viewpoint disturbances.

| Success Rate | ReViWo-BC (Training) | ReViWo-BC (CIP) | ACT (Training) | ACT (CIP) |
|---|---|---|---|---|
| Stage 1 | 100% | 100% | 100% | 100% |
| Stage 2 | 80% | 60% | 60% | 0% |
| Stage 3 | 60% | 50% | 60% | 0% |

Table 1: Results on real world ALOHA robot.

### 4.3 ANALYSIS ON VIEW-INVARIANT REPRESENTATION LEARNING

Previous results demonstrate ReViWo can effectively handle common viewpoint disturbance compared to baseline methods. In this section, we investigate the source of the robustness and analyze the quality of the learned view-invariant representation.

**Analysis on the generated VIR.** We analyze the effectiveness of the learned view-invariant representation by projecting the resulting representations onto a two-dimensional plane using t-SNE (der

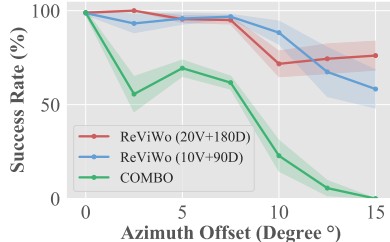

Figure 5: Performance across different levels of viewpoint offsets.

| Method \ Task | | w/ Open-X | w/o Open-X |
|---|---|---|---|
| Door Open | CIP | **42.2** | 22.8 |
| | CSH | 19.4 | **25.0** |
| Drawer Open | CIP | **55.0** | 30.6 |
| | CSH | **37.8** | 25.6 |

Table 2: Ablation study of ReViWo that trains w/ and w/o Open X-Embodiment dataset.

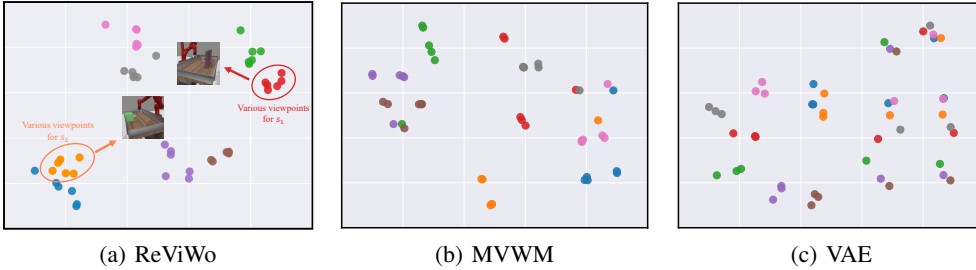

(a) ReViWo       (b) MVWM       (c) VAE

Figure 6: The t-SNE projections of representations generated by various methods. Points with the same color encode six images for the same state but with six different *novel viewpoints* that do not exist in the multi-view datasets. The presented ReViWo representation is generated by VIE.

Maaten et al., 2008). We first sample the state from the Meta-World environment, from which we render a series of images from multiple viewpoints. Subsequently, we encode these images using the VIE, as well as several baseline models for comparison. The t-SNE projections of these encoded images are illustrated in Fig. 6, where each color of points correspond to the representations of the same state from various viewpoints. We observe a notable clustering of VIR points, which are in closer proximity to each other as compared to those of the baseline models. In contrast, the MVWM and VAE representations exhibit greater diversity compared to ReViWo. These results suggest their representations are significantly different across viewpoints, potentially distracting the robust policy execution. This finding indicates the learned VIR is more adept at capturing the essential state information of a task, even in the presence of novel viewpoints.

**Examples of the decoder output.** To investigate whether the trained autoencoder learns meaningful representation, we present the decoded images when VIE and VDE are from different images, as shown in Fig. 7. We have several conclusions from the results. First, VIR correctly focuses on the task state, which is consistent despite changes in viewpoint. For example in each row in Fig. 7, the decoder can generate the images for the same state from two viewpoints. Second, the view-dependent representation indeed captures the property of viewpoint, thus facilitating the generation of the image with the same viewpoint as the viewpoint reference image. We present more examples of decoder output in Appendix F.3. Besides, the trained autoencoder can be applied in different domains (i.e., Meta-world and PandaGym).

## 4.4 ABLATION STUDY AND APPLICABILITY ANALYSIS

**Effect of the Open X-Embodiment data.** To evaluate the impact of Open X-Embodiment data on the efficacy of our proposed method, ReViWo, we conducted a comparative analysis of the model's performance with and without the integration of this data. The results of this analysis are detailed in Tab. 2. The integration of Open X-Embodiment data into ReViWo significantly enhances the model's robustness to variations in viewpoint, as shown in the experiment results. This could be attributed to that the diverse visual representations in the Open X-Embodiment data contribute to a more comprehensive understanding of the world by the model, as well as a better understanding of different viewpoints. These results imply a potential to scale up the ReViWo training for more challenging robotic manipulation. Note that the performance decline on Door Open (CSH) can be attributed to the diverse and unstructured nature of the Open X-Embodiment dataset, which introduces variability that the model struggles to generalize under dynamic conditions.

**Effect of the world model.** We verify the effect of the world model by training a policy with the learned VIR and offline RL method, CQL (Kumar et al., 2020), on the offline RL dataset. Tab.

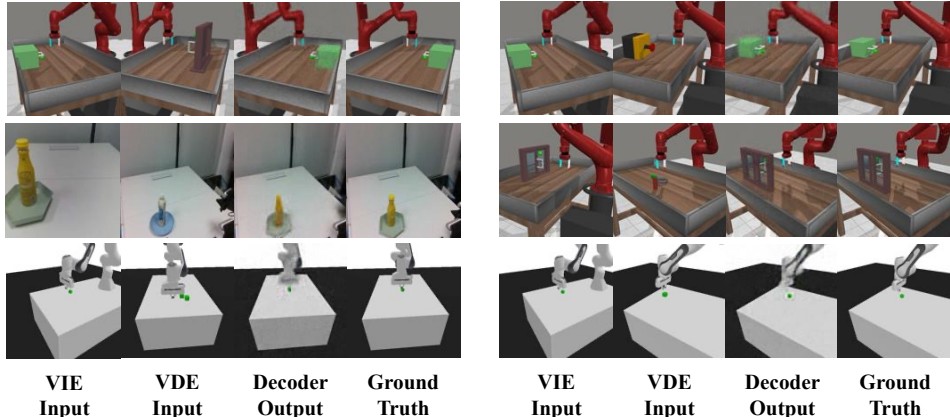

| VIE Input | VDE Input | Decoder Output | Ground Truth | | VIE Input | VDE Input | Decoder Output | Ground Truth |

Figure 7: Examples of the decoder output the corresponding ground truth image. The decoder can generate the images that combine the task state in VIE inputs, and the viewpoint in VDE inputs.

3 presents the performance of ReViWo method with and without world model (WM). The results present a consistent performance enhancement for ReViWo w/ WM, suggesting that the imaginary rollouts generated by the WM contribute to the enhancement of both the robustness and the overall learning efficiency of the ReViWo framework. The reduced performance on Drawer Open (CSH) is likely due to the world model's difficulty in accurately predicting the next state in highly variable and unstable visual conditions, leading to less reliable policy execution. Notably, the ReViWo w/o WM still surpasses the performance of baselines like COMBO (result in Sec. 4). This finding reveals the efficacy of the VIR in facilitating robust control, independent of the world model.

| Task Method | Drawer Open | | | Window Close | | |
|---|---|---|---|---|---|---|
| | Training | CIP | CSH | Training | CIP | CSH |
| w/ WM | **92.8** | **30.6** | 25.6 | **98.9** | **71.4** | **23.3** |
| w/o WM | 60.0 | 28.3 | **27.2** | 97.2 | 68.3 | 17.8 |

Table 3: Performance of ReViWo w/ and w/o world model (WM). We implement ReViWo w/o WM by training policy with CQL and using the learned view-invariant representation.

## 5 CONCLUSION AND LIMITATION

This study explores robust robotic manipulation under viewpoint disturbances. We propose a novel approach, ReViWo, which learns viewpoint-invariant representations that are subsequently leveraged for robotic control. We conduct extensive experiments and demonstrate that ReViWo is capable of being applied to a variety of robotic platforms, enabling robust control under various types of viewpoint disturbance. Despite the promising results, there are still limitations. One limitation of our work is the reliance on the labeled viewpoints. In a practical setting, these labels can be generated by sampling multiple trajectories using fixed-position cameras. However, it is worth exploring to eliminate this dependency and to leverage a broader spectrum of multi-view data from the real world. A potential solution is to train with label-free multi-view data and constrain the information contained within VDR, by information bottleneck (Tishby & Slonim, 2000) technique or by masking portions of the image input to the VDR, thereby preventing the VDR contains too much information. Besides, the experiment scale is limited, in terms of the dataset scale and model size. In future works, we hope to scale up the framework to solve more challenging tasks. For instance, employing a pre-trained model such as CLIP (Radford et al., 2021), and training with more data from Open X-Embodiment and other open multi-view data (Yu et al., 2023). Lastly, currently we implement the world model based on a simple multi-layer perceptron architecture. To deal with robot manipulation tasks with more complex dynamics, it would be beneficial to use more powerful structures to build the world model, e.g., recurrent state-space model (Hafner et al., 2019). We believe these interesting directions are worth further exploration for developing smarter and more robust robots with the support of world models and reinforcement learning.

AUTHORSHIP AND CREDIT ATTRIBUTION

The contributions of the authors are as follows:

- Y. Y.: led the direction, oversaw the project;
- J.-C. P.: designed the method;
- N. T.: proved the theory;
- N. T.: implemented the codes;
- J.-C. P.: designed the experiments;
- K. L.: processed the data;
- N. T., K. L.: conducted experiments;
- J.-C. P., K. L., Y. T.: analyzed the results;
- J.-C. P., X.-Q. C., Y. T., Z.-Y. Z.: wrote the paper;
- Y. Y., M. S., G. N.: revised the paper;
- J.-C. P., G. N.: discussed and wrote the response;
- Y. Y., M. S., G. N.: provided guidance.

All authors discussed and contributed to the final manuscript.

ACKNOWLEDGMENT

This work was supported by the National Science Foundation of China (62495093) and Jiangsu Science Foundation (BK20243039). The authors would like to thank Zhilong Zhang, Junyin Ye and Haoxiang Ren for their assistance in real world deployment, and anonymous reviewers & chairs for their valuable comments.

Masashi Sugiyama was supported by JST ASPIRE Grant Number JPMJAP2405 and by a grant from Apple, Inc. Any views, opinions, findings, and conclusions or recommendations expressed in this material are those of the authors and should not be interpreted as reflecting the views, policies or position, either expressed or implied, of Apple Inc.

ETHICS STATEMENT

In the development and evaluation of ReViWo for robotic manipulation, we have carefully considered the ethical implications of this research, particularly as they pertain to the use of robotic manipulation tasks and artificial intelligence. The proposed involves the collection and use of multi-view data in simulators and Open X-Embodiment datasets and is designed to respect privacy and ensure the security of data. The datasets used do not contain any personal or sensitive information, and all data collection processes comply with relevant legal standards and best practices in research ethics. The potential for bias and discrimination has been addressed by ensuring that the VIR does not inadvertently encode any biased representations of the environment. This is particularly important in maintaining fairness and avoiding any form of discrimination that could arise from biased training data. The research has been conducted with a commitment to research integrity, including thorough documentation and adherence to IRB guidelines where applicable. We recognize that the insights and methods presented in this paper must be applied responsibly, avoiding any potentially harmful applications. The technology developed is intended for beneficial purposes and should not be used in ways that could cause harm or diminish the safety of individuals. The experiment results are reported with the most transparency and accuracy, reflecting our commitment to advancing knowledge in the field of robotic manipulation while upholding the highest ethical standards.

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

# Appendix

## Table of Contents

## A    MATHEMATICAL JUSTIFICATION FOR REPRESENTATION LEARNING

Here we provide a mathematical explanation to justify why ReViWo could separately learn the view-invariant and view-dependent representations from the learning objective in Eq. (3). The intuition is that the disentanglement can be achieved by minimizing the InfoNCE losses, which equals to the contrastive term in Eq. (3).

Denote $V$, $L$ as the random variable of the viewpoint and the scene occupancy, $Z_v$, $Z_l$ as the VDE and VIE. It is obvious that the viewpoint is independent of the occupancy, which means that $I(V; L) = 0$, where $I$ denotes the mutual information. Each RGB image $X$ corresponds to a unique pair $(V, L)$, i.e. $X = X(V, L)$.

During training, we sample a training image batch with size $(N_l, N_v)$ at each step. In this sample matrix, the samples of the same row are sampled from the same state and the samples of the same column are sampled from the same viewpoint.

We use $f(\cdot, \cdot)$ as the similarity metric, and for a certain VDE $z_v$, denote $z_v^P$ its positive sample and $z_v^k$ its negative samples, the InfoNCE loss for VDE is:

$$
\begin{aligned}
L_{\text{nce(VDE)}} &= \sum_{i=1}^{N_l} \sum_{j=1}^{N_v} \left[ -\frac{1}{N_l \cdot N_v} \cdot \log \frac{\exp(f(z_v^{i,j}, z_v^{P_{i,j}}))}{\sum_{k=1}^{N} \exp(f(z_v^{i,j}, z_v^k))} \right] \\
&\approx \mathbb{E}_{v \sim P_v, l \sim P_l, z_v \sim q_v(\cdot|x(v,l))} \left[ -\log \frac{\exp(f(z_v, z_v^P))}{\sum_{k=1}^{N} \exp(f(z_v, z_v^k))} \right] \\
&\approx \mathbb{E}_{v \sim P_v, l \sim P_l, z_v \sim q_v(\cdot|x(v,l))} \left[ -\log \frac{\exp(f(z_v, z_v^P))}{N \cdot \mathbb{E}\left[ \exp(f(z_v, z_v')) \right]} \right] \\
&\approx \mathbb{E}_{v \sim P_v, l \sim P_l, z_v \sim q_v(\cdot|x(v,l)), l^P \sim P_l, z_v^P \sim q_v(\cdot|x(v,l^P))} \left[ -f(z_v, z_v^P) \right] + \\
&\quad \mathbb{E}_{v \sim P_v, l \sim P_l, z_v \sim q_v(\cdot|x(v,l))} \left[ \log(\mathbb{E}_{v' \sim P_v, l' \sim P_l, z_v' \sim q_v(\cdot|x(v',l'))} \left[ \exp(f(z_v, z_v'))) \right]) \right] + \log N \\
&\geq \mathbb{E}_{v \sim P_v, l \sim P_l, z_v \sim q_v(\cdot|x(v,l)), l^P \sim P_l, z_v^P \sim q_v(\cdot|x(v,l^P))} \left[ -f(z_v, z_v^P) \right] + \\
&\quad \mathbb{E}_{v \sim P_v, l \sim P_l, z_v \sim q_v(\cdot|x(v,l)), v' \sim P_v, l' \sim P_l, z_v' \sim q_v(\cdot|x(v',l'))} \left[ f(z_v, z_v') \right] + \log N.
\end{aligned}
\tag{5}
$$

The last inequality in Eq. (5) is derived from Jenson inequality. We set $f(z_v, z_v') = \log P(z_v, z_v') = \log P(z_v) + \log P(z_v'|z_v)$, i.e. the energy function, we have:

$$
\begin{aligned}
L_{nce(VDE)} &\geq \mathbb{E}_{v \sim P_v, l \sim P_l, z_v \sim q_v(\cdot|x(v,l)), l^P \sim P_l, z_v^P \sim q_v(\cdot|x(v,l^P))} \left[ -f(z_v, z_v^P) \right] + \\
&\quad \mathbb{E}_{v \sim P_v, l \sim P_l, z_v \sim q_v(\cdot|x(v,l)), v' \sim P_v, l' \sim P_l, z_v' \sim q_v(\cdot|x(v',l'))} \left[ f(z_v, z_v') \right] + \log N \\
&= \mathbb{E}_{v \sim P_v, l \sim P_l, z_v \sim q_v(\cdot|x(v,l)), l^P \sim P_l, z_v^P \sim q_v(\cdot|x(v,l^P))} \left[ -\log P(z_v) - \log P(z_v^P|v)) \right] + \\
&\quad \mathbb{E}_{v \sim P_v, l \sim P_l, z_v \sim q_v(\cdot|x(v,l)), v' \sim P_v, l' \sim P_l, z_v' \sim q_v(\cdot|x(v',l'))} \left[ \log P(z_v) + \log P(z_v') \right] + \log N \\
&= H(Z_v|v) - H(Z_v) + \log N \\
&= -I(Z_v; V) + \log N.
\end{aligned}
\tag{6}
$$

We justify that $f(z_v, z_v')$ can be represented as $\log P(z_v, z_v')$, which allows us to relate the similarity metric to probabilistic terms. This relationship is crucial for interpreting the InfoNCE loss in terms of entropy and mutual information. Similarly, we obtain:

$$
L_{\text{nce(VIE)}} \geq -I(Z_l; L) + \log N,
\tag{7}
$$

We then prove that the distance metric $d(X, Y)$, defined in terms of mutual information, satisfies the triangle inequality. This metric is used to measure the disentanglement between $Z_v$ and $Z_l$. We denote

$$
d(X, Y) = 1 - \frac{I(X; Y)}{max(H(X), H(Y))},
$$

and

$$A(X_1, X_2, \cdots, X_n) = max(H(X_1), \cdots, H(X_n)).$$

According to triangle inequality, we have:

$$d(V, L) \leq d(Z_v, V) + d(Z_l, L) + d(Z_v, Z_l), \tag{8}$$

which can be simplified as:

$$
\begin{aligned}
I(Z_v; Z_l) &\leq A(Z_v, Z_l) \cdot \left[ 2 - \frac{I(V; Z_v)}{A(V, Z_v)} - \frac{I(L; Z_L)}{A(L, Z_L)} \right] \\
&\leq A(Z_v, Z_l, V, L) \cdot \left[ 2 - \frac{I(V; Z_v)}{A(V, Z_v, L, Z_l)} - \frac{I(L; Z_L)}{A(L, Z_L, V, Z_v)} \right] \\
&= 2 \cdot A(Z_v, Z_l, V, L) - I(V; Z_v) - I(L; Z_l) \\
&\leq 2 \cdot A(Z_v, Z_l, V, L) + L_{\mathrm{nce(VDE)}} + L_{\mathrm{nce(VIE)}} - 2 \log N.
\end{aligned}
\tag{9}
$$

From Eq. (9), we can see that minimizing $A(Z_v, Z_l, V, L)$, $L_{\mathrm{nce(VDE)}}$ and $L_{\mathrm{nce(VDE)}}$ equals to minimizing the upper bound of the mutual information between the view-invariant and view-dependent representations.

Based on the Data Processing Inequality, we show that $A(Z_v, Z_l, V, L)$ is a constant $C$, which represents the maximum entropy among the variables. This conclusion is based on the fact that processing data cannot increase mutual information:

$$
\begin{aligned}
A(Z_v, Z_l, V, L) &= max(H(Z_v), H(Z_l), H(V), H(L)) \\
&= max(H(V), H(L)) \\
&= C.
\end{aligned}
\tag{10}
$$

Finally, we establish a clear connection between the InfoNCE loss and mutual information. Minimizing the InfoNCE losses for $Z_v$ and $Z_l$ corresponds to minimizing the mutual information between these representations, thus facilitating their disentanglement.

## B  RELATED WORK

**Robotic manipulation with visual inputs** has become a prominent field of study, leveraging cameras to capture environmental data and employing learning-based methods, such as reinforcement learning, to train control policies (Zeng et al., 2018; Kalashnikov et al., 2018b; Quillen et al., 2018; Ebert et al., 2018; Pang et al., 2023a;b; 2021). For instance, the work of VPG (Zeng et al., 2018) utilizes the Deep Q-Network algorithm (Mnih et al., 2015) to train a policy for a block grasping task. Similarly, QT-Opt (Kalashnikov et al., 2018a) has demonstrated the feasibility of real-world grasping by introducing a self-supervised, vision-based reinforcement learning framework.

A critical challenge in training policies with visual inputs is the issue of **viewpoint disturbance**, the camera position changes during deployment. Prior research has investigated a variety of methods to mitigate the effects of viewpoint changes. A prevalent technique is domain randomization (Tobin et al., 2017), which augments the training data with synthetic variations to emulate potential environmental disturbances. Another approach is the adoption of unified view representations (Liu et al., 2024; Sermanet et al., 2018; Yin et al., 2022). For instance, RoboUniView (Liu et al., 2024) and TCN (Sermanet et al., 2018) aim to learn a unified representation for images captured from disparate viewpoints. However, the enforcement of viewpoint invariance, which presumes the same information across different viewpoints, necessitates the careful selection of positive and negative data pairs. This is akin to the challenges faced by contrastive learning methods, which often rely on complicated choices with regard to the sampling of such pairs (Saunshi et al., 2019; Seo et al., 2023). Imitation learning methods, such as RT-1 (Brohan et al., 2023), RT-2 (Zitkovich et al., 2023), and RT-X (O'Neill et al., 2024), seek to learn robust control by leveraging extensive manipulation datasets, but this approach inevitably increases data processing and computational demands. While these methods have shown a certain level of success in handling viewpoint disturbances, they fail to isolate view-invariant information, which is essential for robust robotic control Kang (2024), from the viewpoint information. In contrast, this work proposes a novel approach to this challenge.

We introduce an autoencoder framework with dual encoders, designed to decompose view-invariant information, which is shown to be effective in robust robotic control.

**World Models for Robotics Manipulation.** Model-based RL trains a world model that learns the environment dynamics through supervised learning. In robotics, world models are crucial for enabling robots to predict the outcomes of their actions. There are two common ways to utilize the world model in robotics: planning (Alterovitz et al., 2016) and policy learning (Ha & Schmidhuber, 2018). For example, Schmerling et al. (2018) proposes model-based planning for human-machine interaction. Ha & Schmidhuber (2018) introduces the concept of world models in the context of reinforcement learning, demonstrating that compact and efficient representations of the environment can be learned, which in turn can be used to train agents with fewer interactions with the real world.

Due to the challenge in modeling the environment with image as input/output, recent works propose to separate the representation learning and world modeling (Hafner et al., 2020; 2021; 2023). However, these world models seldom consider the viewpoint disturbance. In this work, we follow the representation learning + world modeling framework but attempt to learn a view-invariant world model that is robust to the viewpoint disturbances.

## C  MORE IMPLEMENTATION DETAILS & EXPERIMENT SETUP

### C.1  VIEWPOINTS FOR AUTOENCODER AND RL TRAINING

To construct the dataset to train the autoencoder, we use a camera configuration that encompasses a diverse set of azimuth angles. Specifically, the azimuth angles employed in Metaworld are {8, 11, 17, 20, 28, 35, 136, 144, 148, 152, 159, 180, 194, 195, 198, 219, 324, 333, 339, 351}. Note that these angles are selected to span a 180-degree range, encompassing two distinct 90-degree intervals: [-45, 45] and [135, 225] degrees, respectively. For PandaGym, the azimuth angles are {6, 21, 39, 60, 69, 85, 114, 125, 153, 159, 162, 170, 179, 205, 209, 213, 217, 224, 238, 244} selected from an interval of [0, 270] degrees.

For RL training, we select a distinct azimuth angle of 22.5 degrees for the viewpoint of MetaWorld and 45 degrees for the viewpoint of PandaGym. This angle is set to be different from any of the azimuth angles used for training the autoencoder to ensure diverse and comprehensive learning. This choice of viewpoint is designed to evaluate the generalization capabilities of the RL algorithm by exposing it to a novel viewpoint that is not encountered during the autoencoder training phase. We present the visualization of different viewpoints in Fig. 8.

### C.2  ARCHITECTURE AND OPTIMIZATION DETAILS

The autoencoder architecture is based on the ViT (Dosovitskiy et al., 2021). The architecture of the encoder and decoder are presented in Fig. 9. We first utilize a one-layer CNN to project the input image with size $128 * 128$ into patch embeddings where the patch size is $16 * 16$. We employ a bidirectional transformer structure for both encoder and decoder block, consisting of 8-head self-attention mechanisms and feed-forward networks with layer normalization and residual connections included. The transformer's intermediate embedding dimension is 256 and the dropout rate is set 0.1. For VIE, we utilize a vector quantizer with a codebook size of 512 and an embedding dimension of 64 to obtain the view-invariant encoding. To prevent codebook collapse, we utilized k-means initialization for our vector quantizer. As for VDE, we introduce a simple linear layer to get the view-dependent encoding. The view-invariant encoding and view-dependent encoding are concatenated and fed into the decoder block, after which the decoder output is projected into the image space through a one-layer deconvolution.

### C.3  COMPUTATION

We use 64 CPU cores (AMD EPYC 9654 @ 2.4GHz) and 4 GPUs (NVIDIA GeForce RTX 4090) for our experiments. The software stack employed for our experiments includes Python 3.11 and PyTorch 2.1.0. On average, the training of the autoencoder to the point of convergence on our dataset

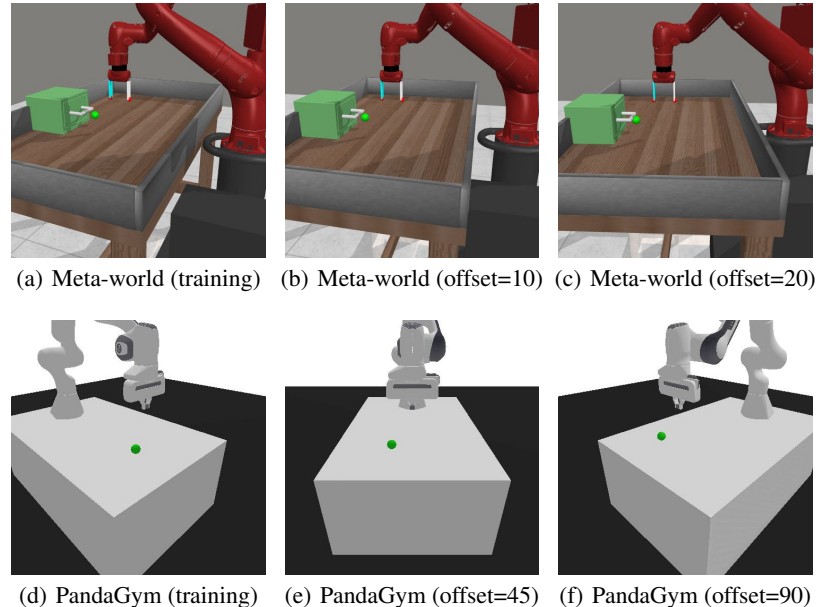

(a) Meta-world (training)  (b) Meta-world (offset=10)  (c) Meta-world (offset=20)

(d) PandaGym (training)  (e) PandaGym (offset=45)  (f) PandaGym (offset=90)

Figure 8: Visualization of the training viewpoint and novel viewpoint for robotics control. In our experiments, Meta-world (offset=10) and PandaGym (offset=90) are utilized for evaluating policy performance under CIP.

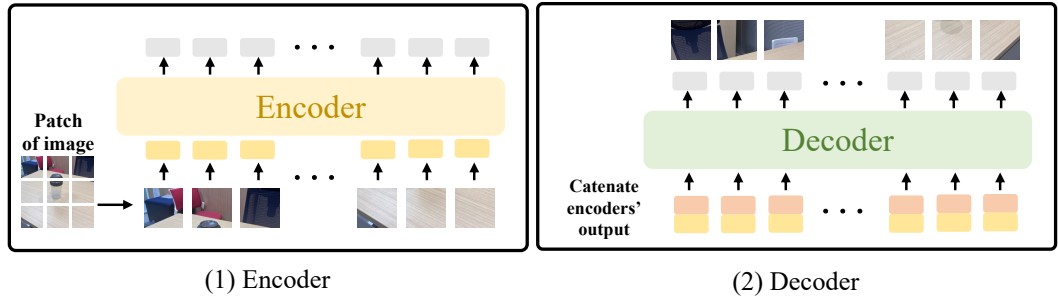

(1) Encoder  (2) Decoder

Figure 9: Network architecture of encoder and decoder of ReViWo.

requires approximately 12 hours. Subsequently, the training of the world model and the policy are completed in an average of 2 hours.

### C.4 HYPERPARAMETERS

The hyper-parameters for implementing ReViWo are presented in Table 4. When implementing baseline methods, we use the same hyper-parameters of offline RL for training the world model and policy.

### C.5 DETAILED DESCRIPTIONS OF BASELINES

**MVWM** (Seo et al., 2023): MVWM is a reinforcement learning framework that trains a multi-view masked autoencoder for representation learning and a world model to solve visual manipulation tasks. The autoencoder consists of a synergistic combination of view-masking: which masks viewpoints at random, and video autoencoding: which reconstructs video frames of both masked and unmasked viewpoints. In Seo et al. (2023), the authors find that the autoencoder effectively learns representations that capture useful information from the current viewpoint but also the cross-view information from different viewpoints. For behavior learning, MVWM learns a world model on

Table 4: Hyper-parameters for training ReViWo and baselines.

| Hyper-parameters | Value |
|---|---|
| Patch Num. | $8 \times 8 = 64$ |
| Embedding dimension for each patch | 256 |
| Encoder output dimension | 64 |
| Enocoder block Num. | 8 |
| Decoder block Num. | 8 |
| Coef. for VQ loss | 0.25 |
| Coef. for VDE contrastive loss | 0.1 |
| Coef. for VIE contrastive loss | 1 |
| Learning rate of authencoder | 3e-5 |
| Optimizer for autoencoder | Adam |
| Learning rate of world model | 1e-3 |
| Optimizer for world model | Adam |
| Learning rate for policy | 1e-4 |
| Optimizer for policy | Adam |
| World model structure | [input dim, 2048, 512, 256, 256, output dim] |
| World model ensemble Num. | 7 |
| Policy structure | [input dim, 2048, 512, 256, 256, output dim] |

frozen representations from either single-view or multi-view data, which is particularly feasible as the autoencoder consists of vision transformer layers that take inputs of variable sizes. Then the actor and critic are trained with imaginary trajectories from the world model. Although MVWM is originally implemented for online RL, it can be easily adapted to offline settings by combining its representation learning technique with offline model-based RL (Agarwal et al., 2020).

**COMBO** (Yu et al., 2021): COMBO is a novel approach in the field of reinforcement learning, designed to effectively learn policies from offline data without further interaction with the environment. It addresses the challenge of distributional shift, where the policy encounters states that are not well-represented in the offline dataset, which can lead to poor performance or even catastrophic failures when deployed. COMBO integrates conservative policy evaluation with model-based planning, using a learned dynamics model to simulate future states and rewards. By combining conservative value estimation and model-based rollouts, COMBO aims to safely optimize policies while mitigating the risks associated with out-of-distribution states. This approach allows for more robust policy learning from static datasets, expanding the applicability of reinforcement learning to scenarios where active data collection is impractical or impossible.

**CQL** (Kumar et al., 2020): CQL is an algorithm developed for offline reinforcement learning, where the goal is to learn effective policies from a fixed dataset without further interaction with the environment. The key challenge in offline reinforcement learning is to avoid overestimation of the action values (Q-values) for state-action pairs not well-represented in the dataset, which can lead to sub-optimal or dangerous policies when executed in the real environment. CQL addresses this issue by introducing a conservative estimation of Q-values during training. It does so by penalizing the Q-values of actions that are not supported by the data and by ensuring that the learned policy does not deviate too much from the behavior policy that generated the dataset. This conservative approach reduces the likelihood of overestimating the Q-values of unseen actions, leading to more reliable policy performance. CQL is particularly useful in scenarios where data collection is expensive or risky, such as robotics, healthcare, or finance, where it is critical to learn from limited data without the opportunity for trial-and-error learning in the actual environment. By focusing on the safe and robust optimization of policies from offline data, CQL represents a significant step forward in the practical application of reinforcement learning.

**BC**: Behavior cloning (BC) is a straightforward method in the realm of robotic learning, where the goal is to mimic expert behavior. BC trains a policy (typically represented by a neural network) to replicate the actions taken by an expert in various states. This is achieved by collecting a dataset of state-action pairs from an expert's demonstrations and then using this dataset to train the policy via supervised learning, treating the problem as a simple function approximation task where the input

is the state and the output is the action. The appeal of BC lies in its simplicity and efficiency, as it does not require reinforcement signals or interaction with the environment during training. Thus, BC has emerged as a popular method employed in the realm of robots, where learning a capable policy by online interaction is challenging and expensive. However, it also has limitations, such as the tendency to accumulate errors due to covariate shift, where the policy encounters states that are not well-represented in the training data, leading to actions that deviate from the experts', and thus to states that are even less well-represented, in a potentially compounding fashion. Despite its limitations, BC can be quite effective for tasks where the expert's policy is easy to capture with supervised learning, and it serves as a foundation for more complex imitation learning algorithms that seek to address its shortcomings.

## D ALGORITHM DESCRIPTION

The practical implementation of ReViWo method is presented in the form of pseudo-code in Algorithm 1.

---

**Algorithm 1** Representation learning for View-invariant World model (ReViWo)

---

**Required**: a multi-view image dataset $\mathcal{O}$, offline control data $\mathcal{D}$ and a empty replay buffer for world model rollout $\tilde{\mathcal{D}}$.
**Output**: the optimized robotic control policy $\pi$.

1: Initialize the autoencoder $p_\phi, q_\phi$, world model $\mathcal{M}_\theta$, reward model $\mathcal{R}_\Theta$ and policy $\pi_\Phi$, where the subscript denotes their parameters.
2: // Training autoencoder
3: **while** training not converge **do**
4:     Sample $(o_{s_1}^{v_1}, o_{s_2}^{v_2}, o_{s_1}^{v_2})$ data from $\mathcal{O}$.
5:     Update $\phi$ based on AE training objective in Eq. (3).
6: **end while**
7: Pre-process visual observation in $\mathcal{D}$ from $s_t$ to $z_{s_t}$.
8: Update the offline control dataset with $(z_{s_t}, a_t, z_{s_{t+1}}, r_t)$.
9: // Training world model
10: **while** AE training not converge **do**
11:     Sample $(s_t, a_t, s_{t+1}, r_t)$ from $\mathcal{D}$.
12:     Update world model parameters $\theta$ with world model training objective in Eq. (4).
13:     Update reward model parameters $\Theta$ with supervised learning on reward data.
14: **end while**
15: // Training policy
16: **while** policy training not converge **do**
17:     Sample in the world model with $\pi_\Phi$ and $\mathcal{R}_\Theta$ to collect $(\tilde{z}_{s_t}, a_t, \tilde{z}_{s_{t+1}}, \tilde{r}_t)$.
18:     Update replay buffer: $\tilde{\mathcal{D}} = \tilde{\mathcal{D}} \cup (\tilde{z}_{s_t}, a_t, \tilde{z}_{s_{t+1}}, \tilde{r}_t)$.
19:     Sample from both $\mathcal{D}$ and $\tilde{\mathcal{D}}$ to update $\Phi$ with COMBO algorithm.
20: **end while**
21: **return** the optimized policy $\pi$.

---

## E DISCUSSION ON THE USAGE OF DATASET WITH VIEW LABELS

In this research, we present a method for robust robotic manipulation that relies on a multi-view dataset that has implicit view labels to train the autoencoder. We would like to discuss the rationality of employing such labels for robust robotic control. The view information is important in visual robotics control. The ability to discriminate between features that are relevant under different viewing conditions is paramount. View labels facilitate this process by guiding the model to associate specific features with the viewpoints they are most relevant to, thereby enhancing the model's ability to generalize these features across a range of similar viewpoints. Some previous works (Liu et al., 2024) also attempt to train with the view label. Different from Liu et al. (2024), we do not explicitly use the view labels like camera parameters, but the multi-view images that are naturally aligned in states or viewpoints.

Besides, in practice, it is not expensive to obtain the viewpoint labels, as we can record multiple manipulation trajectories from some fixed-position cameras. Such view information is not just a facilitator for quicker learning, but also a critical component that imbues the model with a deeper understanding of its operational context. It empowers the model to develop a robust, interpretable, and adaptable framework for robotic manipulation that is essential for real-world applications. Moreover, we demonstrate that ReViWo can be integrated with multi-view data with that is without a view label. Our experiments in Tab. 2 demonstrate the efficacy of such integration.

# F ADDITIONAL RESULTS

## F.1 RESULTS ON MORE TASKS

We conduct experiments on more tasks in the Meta-world environment, including Coffee Button, Faucet Open, and Dial Turn. The experiment results are presented in Fig. F.1.

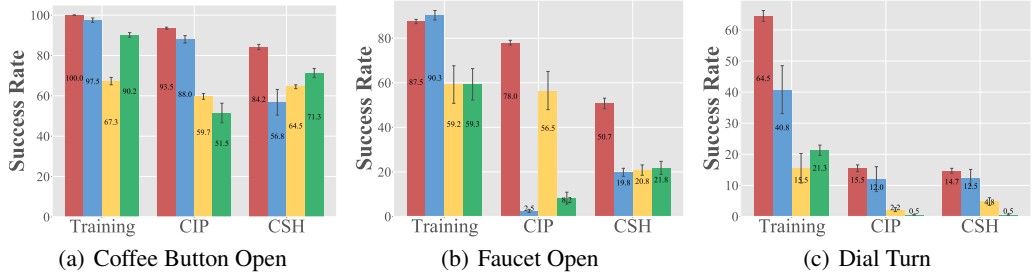

| (a) Coffee Button Open | (b) Faucet Open | (c) Dial Turn |

Figure 10: Performance of Imitation Learning with View-invariant Representation. The x-axis denotes the disturbance type, and the y-axis denotes the average success rate of the last two checkpoints, by evaluation for 30 episodes.

## F.2 IMITATION LEARNING WITH VIEW-INVARIANT REPRESENTATION

Imitation learning is a widely-considered setup in the field of robotics. We further investigate whether the learned VIR can be utilized for imitation learning. To this end, we apply BC with the learned VIR using the offline control data. ReViWo's capability to learn view-invariant representations enables the application of Imitation Learning as an alternative to training a world model for robotic control policy acquisition. Fig. 11 illustrates the success rate of Behavior Cloning utilizing view-invariant representations (RiVi-BC). While RiVi-BC does not surpass ReViWo, it outperforms conventional Behavior Cloning with a VAE encoder. The performance drop of BC is primarily attributed to distributional shift and the accumulation of errors during sequential decision-making. BC is inherently susceptible to these issues, as it relies on imitating actions from a fixed dataset. However, it is noteworthy that when BC is paired with VIR, it outperforms BC with VAE representations. This outcome underscores the robustness of VIR in novel viewpoints and improving overall performance.

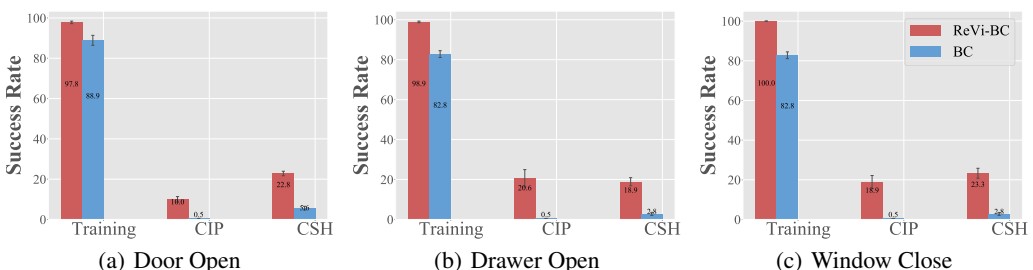

(a) Door Open       (b) Drawer Open       (c) Window Close

Figure 11: Performance of Imitation Learning with View-invariant Representation. The x-axis denotes the disturbance type, and the y-axis denotes the average success rate of the last two checkpoints, by evaluation for 30 episodes.

### F.3 More Examples of Decoder Output

Fig. 12 presents additional examples of the decoder output.

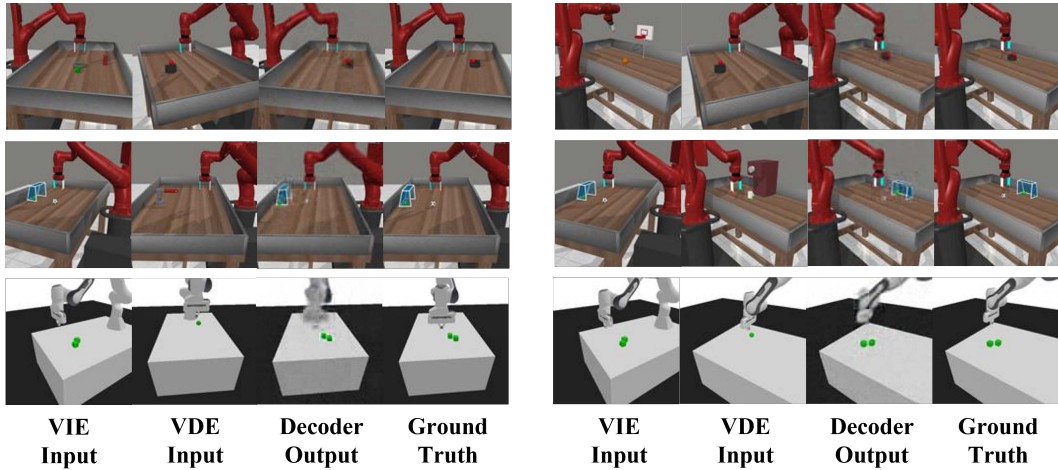

| VIE Input | VDE Input | Decoder Output | Ground Truth | | VIE Input | VDE Input | Decoder Output | Ground Truth |

Figure 12: Additional examples of the decoder output the corresponding ground truth image. The decoder can generate the images that combine the task state in VIE inputs, and the viewpoint in VDE inputs.

