# OpenReview forum: "Learning View-invariant World Models for Visual Robotic Manipulation"
_ICLR.cc/2025/Conference — ICLR 2025 Poster_

### Official Review · Reviewer_pubk · 2024-10-21

**Soundness:** 2
**Presentation:** 3
**Contribution:** 2
**Rating:** 3
**Confidence:** 5

**Summary:**

This paper presents a framework named as ReViWo to learn view-invariant representation (VIR), then uses the VIR as the low-dimensional state representation to train a policy with model-based offline RL (COMBO).


1. ReViWo includes a view-invariant encoder and view-dependent encoder to reconstruct images at different viewpoints by combining VIR with view-dependent information.
2. ReViWo was trained from multi-view simulation data and open-x datasets, and then evaluated on selected tasks on MetaWorld and PandaGym environments.
3. The authors show that ReViWo is robust to viewpoint changes in evaluation.

**Strengths:**

1. It is beneficial to improve the robustness of a robotic policy to camera shaking or viewpoint changes.
2. Finding low-dimensional state representation for high-dimensional visual signals (such as images), and applying existing offline RL methods on that state representation, is an interesting policy structure.
3. The writing is easy to follow.

**Weaknesses:**

**1. (Fundamental Limitation)**

Robotic tasks heavily depends on the understanding of multi-view camera data. For example, it is very common to have a static top camera and a moving on-gripper camera. The focus is on how to leverage information from different views in order to get better performance instead of only using the very limited invariant information. Therefore, by latching on view-invariant information, the proposed ReViWo is limited to only single-view RGB observation without depth, a highly constrained and often impractical setting in robotics  (Note that if we have RGBD, we could reproject the point cloud to multiple views, so RGBD can be considered as multi-view).

Moreover, suppose there is a desk in a single RGBD image, the camera movement can be viewed as the relative movement of the desk as well. This means that a view-invariant representation will lose some ability to sense object layout changes. This is really undesirable.

**2. (Lack of Technical Contribution)**

The method section looks hand-wavy. The authors propose a view-invariant encoder and view-dependent encoder to reconstruct images and learn view-invariant representation. However, the authors did not provide any mathematical guarantee or at least intuition on why the representation can be disentangled that way. It is very likely that the two encoders just work in parallel without having the expected property.

Moreover, the authors mentioned the training of a world model and training a reward model, but in fact, they used the COMBO method to do all these [1]. I am not sure whether COMBO, an offline model-based RL method, can be called a world model because it only predicts the next low-dimensional state. The COMBO framework includes the training of a reward model, so it is not the contribution of this method.

**3. (Lack of Proper Evaluation)**

The authors only evaluate on MetaWorld and PandaGym, two very simple task suites in terms of manipulation diversity, precision, horizon length. Even on these simple task suites, the authors only select 3 tasks from MetaWorld and 1 task from PandaGym, while the MetaWorld has roughly 50 tasks. This evaluation is very insufficient.


[1] COMBO: Conservative Offline Model-Based Policy Optimization. NeurIPs 2021

**Questions:**

(see weakness)

---

> ### Author Response · Authors · 2024-11-22
>
> We are grateful for the time and effort you put into reviewing our work. Below we answer each of your concern and question.
>
> > Q1 (**Fundamental Limitation**): The proposed ReViWo is limited to only single-view RGB observation without depth, a highly constrained and often impractical setting in robotics (Note that if we have RGBD, we could reproject the point cloud to multiple views).
>
> A1: This is a good point. We clarify that while RGBD data can be reprojected to multiple views, it has some limitations. This projection process utilizes the depth information, which is often noisy and can vary significantly with changes in lighting and surface properties, making it challenging to maintain consistent across viewpoints. Additionally, it relies on precise camera calibration (which may not be obtained when viewpoint changes) to project RGBD data to point cloud.  **Therefore, we suggest that an effective learning method is still required to robustly extract robust representation from the visual input**. Towards this direction, ReViWo method provides a potential solution by focusing on robust representation learning.
>
> We acknowledge that single-view observation may be impractical in some scenarios. Hence, we have conducted real world experiments using three-view observations (two third-person cameras and one first-person camera). Please refer to A4 for more details.
>
> > Q2 (**Lack of Technical Contribution**): the authors did not provide any mathematical guarantee or at least intuition on why the representation can be disentangled that way. It is very likely that the two encoders just work in parallel without having the expected property.
>
> A2: We appreciate your concern. We would like to clarify that the two encoders do not simply work in parallel; rather, they learn to capture different types of information through the contrastive term. We now provide a preliminary mathematical justification that the disentanglement of the view-invariant and view-dependent representations can be achieved by minimizing the InfoNCE losses, which we use to implement the contrastive term in Eq. (3). The intuition is that the InfoNCE loss can bound the mutual information between the VIE and VDE representations. Please refer to Appendix F in the revised paper for detailed proofs.
>
> > Q3: The COMBO framework includes the training of a reward model, so it is not the contribution of this method.
>
> A3: We would like to clarify that our primary contribution lies in the novel approach of separately learning the view-invariant and view-dependent information from observations to address the challenges of viewpoint disturbances. Our approach represents an advancement over the standard learning framework that uses a single information extractor to handle viewpoint disturbances. Therefore, we have only briefly described the world modeling aspect.
>
> > Q4 (**Lack of Proper Evaluation**): The authors only select 3 tasks from MetaWorld and 1 task from PandaGym, while the MetaWorld has roughly 50 tasks. This evaluation is very insufficient.
>
> A4: We acknowledge the concern regarding the evaluation scope. To mitigate the concern of task bias, we have included an experiment in real world. This experiment involves a long-term manipulation task involving three stages: reaching for a bottle (stage 1), grasping it (stage 2), and placing it on a plate (stage 3). We collected a dataset of 128 trajectories using three cameras (two thrid-person cameras and one gripper camera). The dataset includes 10 distinct types of bottles and plates, randomly placed within the operational area during data collection. We implement ReViWo-BC as follow: training VIE on the real world images and conducting BC on the control data with VIR. In this experiments, the policy inputs with all three camera images. To construct viewpoint disturbances, we set the azimuth=+15 and the elevation=-15 on one of the third-person cameras. We evaluate the method on both training and CIP viewpoint for 10 trajectories with random initialization. Experimental results are as follow:
>
> | Success Rate| Stage 1 | Stage 2 | Stage 3 |
> | -------------------- |-------| ------- | ------- |
> | ReViWo-BC (Training) | 100% | 80%| 60%|
> | ReViWo-BC (CIP) | 100%| 60%| 50%|
>
> The results indicate that the proposed method maintains robust control when subjected to novel viewpoints in a real-world setting. This result serves as a preliminary evidance of the applicability of ReViWo method. Refer to [this link](https://anonymous.4open.science/r/ReViWo-448B/assets/real_world.gif) for the real world deployment video. We believe this real world experiment is an important supplement to the evaluation.
>
> [**UPDATE 29 Nov.**] We have extended the evaluations to 50 Meta-World tasks. Please refer to [our response](https://openreview.net/forum?id=vJwjWyt4Ed&noteId=oxChgwurah) to Reviewer Czra for details.
>
> ------
> We hope that these responses can address your concerns and questions. If you had any further concerns, please let us know.

---

> ### Author Response · Authors · 2024-12-02
>
> Hi reviewer pubk. We wanted to follow up to see if the response addresses your concerns. If you have any further questions, please let us know. Thank you again!

---

### Official Review · Reviewer_Hd4G · 2024-10-31

**Soundness:** 3
**Presentation:** 3
**Contribution:** 2
**Rating:** 6
**Confidence:** 2

**Summary:**

This paper presents ReViWo (Representation learning for View-invariant World model), a novel approach addressing the challenge of viewpoint changes in robotic manipulation tasks. Traditional methods struggle with performance degradation under varying camera angles; ReViWo overcomes this by leveraging multi-view data to learn robust representations for control under viewpoint disturbances. Using an autoencoder framework, ReViWo combines view-invariant and view-dependent representations, trained on both labeled multi-view simulator data and the Open X-Embodiment dataset (without view labels). Tested in Meta-world and PandaGym environments, ReViWo outperforms baseline methods, maintaining stable performance in scenarios with novel camera angles and frequent camera shaking. These results validate ReViWo’s effectiveness in providing robust, task-relevant representations across diverse viewpoints.

**Strengths:**

View variation is a very practical problem in robotic manipulation. This paper provides good experiments, and leverages Open X-Embodiedment, leveraging this dataset is a promising direction is robotics. Also, the manipulation video in the link is cool.

**Weaknesses:**

1. The paper shows application in simulation. However, in other specific real world scenarios, there is not enough multi-view data for training. So, I recommend authors to demonstrate how the study can work in the real world, and show real-world evaluation results.

2. This paper only shows results in some simple tasks. Will this method work in more diverse tasks? For example, manipulating deformable objects like folding cloth. These experiments will improve the significance of this paper in robotics field.

**Questions:**

1. To my knowledge, the output of VAE is blurred. The multi-step world model will enlarge the blurring problem. Could you please how this problem affects this performance? Also, I will appreciate if the author could release the structure of VAE for reproductibilty.

2. Also, will the vae have some errors? For example, will the generated object have different geometries?

2. See weakness.

---

> ### Author Response · Authors · 2024-11-22
>
> Thank you for your time and valuable comments. We have taken every comment into consideration and conducted real world experiments. Please find the response below.
>
> > Q1:  In other specific real world scenarios, there is not enough multi-view data for training. So, I recommend authors to demonstrate how the study can work in the real world, and show real-world evaluation results && Will this method work in more diverse tasks?
>
> A1: We agree with you at this point. We include a real world experiment, which considers a long-term manipulation task that involves three stages: reaching for a bottle (stage 1), grasping it (stage 2), and then placing it on a plate (stage 3). We collected a dataset of 128 trajectories using three cameras (two thrid-person cameras and one gripper camera). The dataset includes 10 distinct types of bottles and plates, which were placed randomly within the operational area during data collection. We implement ReViWo-BC as follow: training VIE on the real world images and conducting behavior cloning on the control data with VIE representation. In this experiments, the policy inputs with all three camera images. To simulate viewpoint disturbances, we adjusted the azimuth by +15 degrees and the elevation by -15 degrees on one of the third-person cameras. We evaluate the method on both training and CIP viewpoint for 10 trajectories with random initalization. Experimental results are as follow:
>
> | Success Rate         | Stage 1 | Stage 2 | Stage 3 |
> | -------------------- | ------- | ------- | ------- |
> | ReViWo-BC (Training) | 100%    | 80%     | 60%     |
> | ReViWo-BC (CIP)      | 100%    | 60%     | 50%     |
>
> The results indicate that the proposed method maintains robust control when subjected to novel viewpoints in a real-world setting. This result serves as a preliminary evidance of the applicability of ReViWo method. Refer to [this like](https://anonymous.4open.science/r/ReViWo-448B/assets/real_world.gif) for the real world deployment video. We believe this real world experiment is a important supplement to the evaluation.
>
> We also include evaluation results on additional tasks in Appendix E.1. These evaluations demonstrate ReViWo's effectiveness across a broader range of tasks, providing further evidence of its robustness and versatility.
>
>
> > Q2: The output of VAE is blurred. The multi-step world model will enlarge the blurring problem. Could you please how this problem affects this performance? Will the vae have some errors? For example, will the generated object have different geometries?
>
> A2: We agree that VAE outputs can be blurred. To address this, we use VQ-VAE based on the ViT structure, which has shown strong capabilities in image reconstruction. We present an example of the world model inference for 32 timesteps at [this link](https://anonymous.4open.science/r/ReViWo-448B/assets/WM_novel.jpg), demonstrating the effectiveness of our multi-step world model. It is true that image reconstruction may be blurred when processing small-scale objects, such as a ball or a point-like feature. Integrating well-recognized image reconstruction techniques, such as deblurring, could further enhance representation learning.
>
> > Q3: I will appreciate if the author could release the structure of VAE for reproducibility.
>
> A3: We present the structure of encoder/decoder in Figure 9 in Appendix A.2. We first utilize a one-layer CNN to project the input image with size 128 ∗128 into patch embeddings where the patch size is 16 ∗16. We employ a bidirectional transformer structure for both encoder and decoder block, consisting of 8-head self- attention mechanisms and feed-forward networks with layer normalization and residual connections included. The transformer’s intermediate embedding dimension is 256 and the dropout rate is set 0.1. For VIE, we utilize a vector quantizer with a codebook size of 512 and an embedding dimension of 64 to obtain the view-invariant encoding. To prevent codebook collapse, we utilized k-means initialization for our vector quantizer. As for VDE, we introduce a simple linear layer to get the view-dependent encoding. The view-invariant encoding and view-dependent encoding are concatenated and fed into the decoder block, after which the decoder output is projected into the image space through a one-layer deconvolution.
>
> -------
>
> We hope that our response has addressed your concern and questions satisfactorily. If you had any further concerns, we are glad for discussion.

---

> > ### Comment · Reviewer_Hd4G · 2024-11-29
> >
> > I thank the detailed response from the author, and have updated the rating.

---

> > > ### Author Response · Authors · 2024-11-29
> > >
> > > Thank you for your feedback and raising the rating. We appreciate your time and contributions to improve this work.

---

### Official Review · Reviewer_Czra · 2024-10-31

**Soundness:** 3
**Presentation:** 3
**Contribution:** 4
**Rating:** 6
**Confidence:** 3

**Summary:**

The draft proposes and evaluates a new approach for learning a view-invariant encoding and world model, in the context of learning vision-based robotic manipulation.

**Strengths:**

The problem addressed by the draft, view-invariance or at least viewpoint-robustness of learnt robotic manipulation policies, and more generally learning of view-invariant world models, is extremely important and challenging.
The solution proposed by authors for learning a view-invariant representation, which consists in learning a distangling of view-invariant encoding and view-dependant encoding, is appealingly elegant and seems rather original. It also has the interest as a by-product of enabling generation of an arbitrary (?) new view-viewpoint for a given system state, and conversely.
The experiments conducted on Meta-world and Panda-gym reported in the paper are rather convincing regarding the ability of the proposed approach to bring significant viewpoint-robustness (figures 4 and 5), and to learn a relatively view-invariant embedding (figure 6).

**Weaknesses:**

The main weaknesses of the paper are the following:
 - while the "baselines for comparison" include MVWM, no comparison is conducted with other important very-related works mentioned: RT-X series works and RoboUniView
- the experiments are conducted on a quite small number of tasks (3 out of 50 on Meta-World), and only one (!) in Panda-Gym ; this raises some doubt on possible "cherry-picking" approach for choosing these tasks

**Questions:**

The values of the \lambda_1 and \lambda_2 coefficients for respectively the VQ-term and contrastive-term in the training objective (equation 3) are listed in Table 3 of the annex, but authors provide absolutely no clue on how these values were chosen, and do not analyze how critical these values might be for the result (my intuition is that they should have a significant impact). Could authors clarify that, and ideally provide at least a minimal quantitative analysis/assesment of how important it could be ?

In second paragraph of §4.2, authors write that their proposed method "consistently surpasses the baselines", but on figure 4 it can be seen that for task Window-Close, MVWM has much higher success rate (~85%) than ReWiVo on the CIP case, and slighltly higher succes rate on the CSH case. Authors should NOT "over-claim" in their text compared to figure, and should comment on this lesser performance of their method on this task.
Furthermore, the presentation of figure 4 with "skipping" the 40%-80% part of the y axis is somewhat misleading: this must be corrected, possibly by using higher plots that do NOT skip the 40%-80% range.

Regarding the use of Open X-embodiment data, ablation study reported table 1 shows it has a quite significant impact ; however, authors mention on line 227 that they "introduce a weigthing factor in the loss calculation for these unlabeled data", but it seems they provide no information whatsoever about what value is this weighting and how critical this value could be for the outcome.
Also, it appears in table 1 that inclusion of this extra data actually *degrade* result in the CSH case for Door-Open task ; author should comment on that.

Examples of the decoder output shown in figure 7 are impressively similar to the ground truth, but are those examples on *test* data or on some of the training data ??
Furthermode, since the success rate falls from over 90% on training data down to below 40% on test data of the 3 first tasks (a) (b) and (c) on figure 4, there must be a significant number of cases for which the view-invariance does not work so well --> authors should show and comment some of these failure cases, to allow readers to qualitatively evaluate what happens when results are not as good as in current figure 7...

In last part of §4.4, authors write that inclusion of their World Model (WM) "present a consistent performance enhancement", while actually in table 2 providing some ablation on the impact of their world model component, WM appears to slightly *degrade* result for CSH case of Drawer-Open task ; author should comment on that.

---

> ### Author Response · Authors · 2024-11-22
> **Author Response (Part 1/2)**
>
> Thanks for your positive comments and careful comments on the paper. Below we address each of your concerns and questions.
>
> > Q1: The main weaknesses of the paper are the following.
>
> We have conducted experiments with real world robot and more baseline in response to your concerns.
>
> **Comment 1:** while the "baselines for comparison" include MVWM, no comparison is conducted with other important very-related works mentioned: RT-X series works and RoboUniView
>
> **Response 1:** We appreciate the feedback and have now included comparative experiments with the RT-X series. It is important to note that RoboUniView necessitates camera parameters for learning representations, which differs from the experimental setup used for ReViWo. The RT-X series is designed to train policies using extensive datasets to achieve generalization across novel viewpoints. For a fair comparison, we adapted RT-X to utilize multi-view data, aligning it with our experimental conditions. The results of these experiments are presented in the following table:
>
> | Success Rate | RT-X (Training) | ReViWo (Training) |  RT-X (CIP) | ReViWo (CIP) |  RT-X (CSH) | ReViWo (CSH)  |
> | ------------ | --------------- | ----------------- | ---------- | ------------ | ---------- | ------------ |
> | Door Open    | 89.2            | 91.1              | 0.0        | 42.2         | 1.7        | 19.4         |
> | Window Close | 89.3            | 98.9              | 0.0        | 71.1        | 0.0        | 23.3         |
>
> The results illustrate that ReViWo outperforms RT-X in both training success rates and in generalization to new instances and scenes. The could be attributed that RT-X relies on sufficient amounts of data to generalize novel viewpoints, while it is challenging to collect sufficient amount of control data. This result further demonstrates the robustness of ReViWo in viewpoint disturbance.
>
> **Comment 2:** the experiments are conducted on a quite small number of tasks (3 out of 50 on Meta-World), and only one (!) in Panda-Gym ; this raises some doubt on possible "cherry-picking" approach for choosing these tasks
>
> **Response 2:** To clarify, in addition to the initially reported tasks, we also conduct experiments on three additional tasks from Meta-World, with detailed results presented in Appendix E.1. To further mitigate the concern of task selection bias, we have included an experiment in a real-world setting. This experiment involves a long-term manipulation task that involves three stages: reaching for a bottle (stage 1), grasping it (stage 2), and then placing it on a plate (stage 3). We collected a dataset of 128 trajectories using three cameras (two thrid-person cameras and one gripper camera). The dataset includes 10 distinct types of bottles and plates, which were placed randomly within the operational area during data collection. We implement ReViWo-BC as follow: training VIE on the real world images and conducting behavior cloning on the control data with VIE representation. In this experiments, the policy inputs with all three camera images. To simulate viewpoint disturbances, we adjusted the azimuth by +15 degrees and the elevation by -15 degrees on one of the third-person cameras. We evaluate the method on both training and CIP viewpoint for 10 trajectories with random initalization. Experimental results are as follow:
>
> | Success Rate | Stage 1 | Stage 2 | Stage 3 |
> | -------------------- | ------- | ------- | ------- |
> | ReViWo-BC (Training) | 100%    | 80%  | 60%     |
> | ReViWo-BC (CIP)  | 100% | 60% | 50% |
>
> The results indicate that the proposed method maintains robust control when subjected to novel viewpoints in a real-world setting. This result serves as a preliminary evidance of the applicability of ReViWo method. Refer to [this link](https://anonymous.4open.science/r/ReViWo-448B/assets/real_world.gif) for the real world deployment video. We believe this real world experiment is a important supplement to the evaluation.
>
> > Q2: The values of the \lambda_1 and \lambda_2 coefficients for respectively the VQ-term and contrastive-term in the training objective (equation 3) are listed in Table 3 of the annex, but authors provide absolutely no clue on how these values were chosen
>
> A2: Upon investigation, we observe that the VQ loss is relatively insensitive to coefficient variations, with respect to the fidelity of image reconstruction. Consequently, we assign a value of 0.25 to the VQ coefficient for our experiments. In contrast, the coefficient for the contrastive term significantly influences the results. We determine the coefficient for the VDE contrastive term, $\lambda_{VDE}$, to be 0.1, acknowledging that view-dependent encodings may share attributes such as background elements. Conversely, for the VIE contrastive term, $\lambda_{VIE}$, we select a higher value of 0.5 to account for the greater variability in features that can occur due to changes in viewpoint.

---

> ### Author Response · Authors · 2024-11-22
> **Author Response (Part 2/2)**
>
> > Q3: Authors should NOT "over-claim" in their text compared to figure, and should comment on this lesser performance of their method on this task.
>
> A3: Sorry for the improper expression, and we will revise our manuscript to accurately reflect these results and avoid any over-claiming. We suggest that the performance advantage of MVWM on Window Close is because it learns visual representation with sequential modelling, which could be benefitial when dealling with some certain viewpoint disturbance.
>
> > Q4: Authors mention on line 227 that they "introduce a weigthing factor in the loss calculation for these unlabeled data", but it seems they provide no information whatsoever about what value is this weighting and how critical this value could be for the outcome.
>
> A4: We apologize for the oversight. The weighting factor introduced in the loss calculation for unlabeled data is set to 0.5. This value was chosen based on preliminary experiments to balance the influence of the collect data and Open X-Embodiment data, ensuring that the model effectively learns from both. We would like to include a detailed analysis in the revised manuscript.
>
> > Q5: Examples of the decoder output shown in figure 7 are impressively similar to the ground truth, but are those examples on *test* data or on some of the training data; there must be a significant number of cases for which the view-invariance does not work so well --> authors should show and comment some of these failure cases
>
> A5: This is a good point. The examples in Figure 7 are from the training data. For a broader perspective, additional examples, particularly those involving novel viewpoints and real world images, are available at [link1](https://anonymous.4open.science/r/ReViWo-448B/assets/metaworld_flaw.png) and [link2](https://anonymous.4open.science/r/ReViWo-448B/assets/real_world_reconstruct.png). We find that in general, the image reconstruction maybe blurred for reconstructing some object details, but preserves the overall shape. It could be benefitial to integrate the image reconstruction techniques, such as deblurring, for better representation learning. We would like to update the paper to present and discuss the additional results.
>
> > Q6: It appears in table 1 that inclusion of this extra data actually *degrade* result in the CSH case for Door-Open task ; WM appears to slightly *degrade* result for CSH case of Drawer-Open task. Author should comment on that.
>
> A6: When integrating Open X-Embodiment data, the performance decline can be attributed to the diverse and unstructured nature of the Open X-Embodiment dataset, which introduces variability that the model struggles to generalize under dynamic conditions. Similarly, the reduced performance with the integration of the world model is likely due to the world model's difficulty in accurately predicting the next state in highly variable and unstable visual conditions, leading to less reliable policy execution. Even that, we emphasize that the integration of Open X-Embodiment data and world model overall brings a positive effect, on most of the evaluation tasks. We would add the discussion to the paper.
>
> ------
>
> We hope the response has addressed your concerns. But if you have any further questions, please let us know.

---

> > ### Comment · Reviewer_Czra · 2024-11-26
> >
> > We thank authors for taking into account several of my suggestions, including experimental ones.
> > I still think, even with the newly reported experiments and the (relatively clumsy) real-world experiment, that for being definitely convincing on this very interesting topic, it would be highly desirable to base the evaluation on a much larger set of tasks (ideally ALL MetaWorld tasks and all panda-gym as well).
> > In summary your work seems very promising, but for ensuring that experimental results reflects some generality, the exact same algorithm should be confronted to a variety of tasks sufficient to catch specific failure cases.
> > I maintain my score of 6, to be considered as an incitation to experiment on more types of tasks rather than publishing maybe faster but on weaker grounds. Depending on the scorings by other reviewers, a final decision like "complete more experiments before publish" can be as acceptable as another decision like "despite a limited number of experiments, this seems a promising research trask".

---

> > > ### Author Response · Authors · 2024-12-02
> > >
> > > Hi reviewer Czra. We want to follow up to see if the response addresses your concerns. If you have any further questions about the evaluation diversity, please let us know. Thank you again!

---

> ### Author Response · Authors · 2024-11-29
> **Results on Full Meta-world Tasks**
>
> Thank you for your detailed comments. We clarify that our existing experiments primarily focus on the task diversity in terms of environment and robotics platform variations. However, we understand your concerns regarding the scope of experimental evaluations and the potential for 'cherry-picking' results. We would like to advance this research and conduct experiments on all 50 Meta-World tasks. Note that in these experiments, the VIE is only trained on 17 tasks due to the time consuming of training on all tasks. For each task, we collect 100 trajectories of control data for offline RL training. We present the average success rate on these tasks as follow:
>
>
> | Success rate | Training | CIP  | CSH  |
> | ------------ | -------- | ---- | ---- |
> | ReViWo       | **61.4**        | **28.1**    | **24.0**    |
> | COMBO        | 45.2        | 15.3    | 18.2    |
>
> The results demonstrate that **ReViWo maintains superior performance across a broader range of visual tasks, and presents less performance drop under novel viewpoints**, compare to baseline offline RL method. We also observe that the success rate is not very high. This could be attributed that the encoder is only trained on partial tasks (17 out of 50), and we only use 100 trajectories of control data (in main experiments, we use 400) in these experiments.
>
> We believe that these additional findings significantly strengthen the validity of evaluations. We will evaluate other baselines such as MVWM for further comparison. Thank you once again for your encouragement and suggestions. We are glad to discuss if you had any further concerns.

---

### Official Review · Reviewer_juRH · 2024-11-04

**Soundness:** 3
**Presentation:** 3
**Contribution:** 3
**Rating:** 6
**Confidence:** 3

**Summary:**

This study investigates robust robotic manipulation in the presence of camera viewpoints disturbances. It develops viewpoint-invariant representations learning methods with a VAE-like objective. The learned viewpoint-invariant representations are subsequently utilized for robotic control. The experimental results on two simulation environments demonstrate the enhanced robustness across two types of viewpoint disturbances.

**Strengths:**

1. The approach for learning viewpoint-invariant representations is quite novel. This paper employs a view-invariant encoder and a view-dependent encoder, which take two images from different viewpoints as input. The features encoded by these two branches are then processed through a decoder, utilizing a VAE-like learning objective to decompose view-invariant and view-dependent information.
2. Both the comparative experiments and the ablation studies are thorough.

**Weaknesses:**

1. The experiments are conducted on two simulation environments only, and the effectiveness of ReViWo on real-world robots remains unvalidated.
2. From Figure 4, even when applying the proposed algorithm ReViWo in this paper, the success rate still significantly declines under disturbances caused by Camera Installation Position (CIP) and Camera Shaking (CSH). This indicates that the enhancement in robustness to viewpoint variations achieved by this method is limited.
3. The representation of Figure 4 is quite misleading. It is recommended to revise it.

**Questions:**

1. Is the proposed algorithm's performance under viewpoint disturbances stable across different tasks, and does it remain consistent outside the simulation tasks involved in this paper?
2. Why does the integration of Open X-Embodiment data into ReViWo lead to a decline in model performance under Camera Shaking (CSH) for the Door Open task? Additionally, why does the integration of the world model into ReViWo result in reduced performance under Camera Shaking (CSH) for the Drawer Open task?

---

> ### Author Response · Authors · 2024-11-22
>
> Thank you for carefully reviewing our paper and valuable comments. We have taken every comment into consideration. Please find the response below.
>
> > Q1: The experiments are conducted on two simulation environments only, and the effectiveness of ReViWo on real-world robots remains unvalidated && does it remain consistent outside the simulation tasks involved in this paper
>
> A1: We understand your concern about real world application. Now we have conducted real world experiments. We consider a long-term manipulation task that involves three stages: reaching for a bottle (stage 1), grasping it (stage 2), and then placing it on a plate (stage 3). We collected a dataset of 128 trajectories using three cameras (two thrid-person cameras and one gripper camera). The dataset includes 10 distinct types of bottles and plates, which were placed randomly within the operational area during data collection. We implement ReViWo-BC as follow: training VIE on the real world images and conducting behavior cloning on the control data with VIE representation. In this experiments, the policy inputs with all three camera images. To simulate viewpoint disturbances, we adjusted the azimuth by +15 degrees and the elevation by -15 degrees on one of the third-person cameras. We evaluate the method on both training and CIP viewpoint for 10 trajectories with random initalization. Experimental results are as follow:
>
> | Success Rate         | Stage 1 | Stage 2 | Stage 3 |
> | -------------------- | ------- | ------- | ------- |
> | ReViWo-BC (Training) | 100%    | 80%     | 60%     |
> | ReViWo-BC (CIP)      | 100%    | 60%     | 50%     |
>
> The results indicate that the proposed method maintains robust control when subjected to novel viewpoints in a real-world setting. This result serves as a preliminary evidance of the applicability of ReViWo method. Refer to [this link](https://anonymous.4open.science/r/ReViWo-448B/assets/real_world.gif) for the real world deployment video. We believe this real world experiment is a important supplement to the evaluation.
>
> > Q2: From Figure 4, even when applying the proposed algorithm ReViWo in this paper, the success rate still significantly declines under disturbances caused by Camera Installation Position (CIP) and Camera Shaking (CSH). This indicates that the enhancement in robustness to viewpoint variations achieved by this method is limited.
>
> A2: While robustness improvement is limited, it is important to note that ReViWo still outperforms representative baseline methods under the same conditions, demonstrating improvement in robustness. The observed decline highlights areas for future enhancement, and the experiments were designed to test the algorithm under severe conditions, with more favorable results under normal conditions. Future work could focus on refining the model to better handle extreme viewpoint variations.
>
> > Q3: Why does the integration of Open X-Embodiment data into ReViWo lead to a decline in model performance under CSH for the Door Open task? Additionally, why does the integration of the world model into ReViWo result in reduced performance under CSH for the Drawer Open task?
>
> A3: When integrating Open X-Embodiment data, the performance decline can be attributed to the diverse and unstructured nature of the Open X-Embodiment dataset, which introduces variability that the model struggles to generalize under dynamic conditions. Similarly, the reduced performance with the integration of the world model is likely due to the world model's difficulty in accurately predicting the next state in highly variable and unstable visual conditions, leading to less reliable policy execution. Even that, we emphasize that the integration of Open X-Embodiment data and world model overall brings a positive effect, on most of the evaluation tasks. We would add the discussion to the paper.
>
> ------
>
> We hope that our response has addressed your concerns, but if we missed anything please let us know.

---

> > ### Comment · Reviewer_juRH · 2024-12-02
> >
> > I agree with Reviewer YacQ's comments that the supplementary real-world experiments are limited to a basic pick-and-place task, which lacks complexity.
> >
> > Additionally, my main concern is about whether position generalization is considered during the evaluation of the real-world experiments. The author noted that ten distinct types of bottles and plates were randomly placed within the operational area during data collection. However, it was not specified whether this randomness was maintained during the evaluation phase.
> >
> > In the real-world experiments, are robot proprioceptive states incorporated? When proprioceptive states are involved and the area of object placement is limited, there's a possibility that the policy could depend more heavily on proprioceptive inputs rather than visual observations. If true, the experimental results on viewpoint generalization become less credible.

---

> > > ### Author Response · Authors · 2024-12-02
> > >
> > > Thanks for your follow-up comments. We answer your questions as follow:
> > >
> > > > Q1: my main concern is about whether position generalization is considered during the evaluation of the real-world experiments.
> > >
> > > A1: Sorry for not clearly specifying the evaluation setting. During the evaluation phase, the initialization randomness was also maintained. This included bottle position from 15cm x 15cm area, plate position from 20cm x 20cm area, and 10 different bottle/plate types to further challenge and evaluate the generalization capabilities of the model.
> > >
> > > > Q2: In the real-world experiments, are robot proprioceptive states incorporated?
> > >
> > > A2: Both ReViWo-BC and the baseline ACT method use the joint position (qpos). We clarify this information is not sufficient for learning a viewpoint-robust policy, as ACT method fails to perform under viewpoint disturbance.
> > >
> > > > Q3: I agree with Reviewer YacQ's comments that the supplementary real-world experiments are limited to a basic pick-and-place task, which lacks complexity.
> > >
> > > A3: We appreciate for your attention to the feedback from other reviewers. It is worth noting that Reviewer YacQ has increased the rating after we presented the [most recent results](https://openreview.net/forum?id=vJwjWyt4Ed&noteId=IzixRs2wMz). We interpret this as an implicit acknowledgement of the latest results to address the raised concerns.
> > >
> > > ------
> > >
> > > We hope our clarification could address your concern. We are happy for further discussions.

---

> > > > ### Comment · Reviewer_juRH · 2024-12-03
> > > >
> > > > Thanks to the author for the responses and the efforts during the rebuttal stage. After comprehensive consideration, I decide to maintain the original score.

---

> > > > > ### Author Response · Authors · 2024-12-03
> > > > >
> > > > > Thanks for your response and positive assessment. We appreciate your time and effort in improving this work.

---

> ### Author Response · Authors · 2024-12-02
>
> Hi reviewer juRH. We want to follow up to see if the response addresses your concerns. If you have any further questions, please let us know. Thank you again!

---

### Official Review · Reviewer_YacQ · 2024-11-05

**Soundness:** 2
**Presentation:** 3
**Contribution:** 2
**Rating:** 6
**Confidence:** 4

**Summary:**

The goal of this paper is to train world models for robot manipulation
that are robust to changes in the position of the camera observing the
manipulator and the environment. This is a very relevant problem,
since cameras frequently move between training sessions and
deployment. To achieve this, the authors introduce a training setup in
which a combination of two VQ-VAE encoders is trained on a multi-view
dataset; one of the encoders, the VIE, learns to encode the
view-independent setting and arrangement of the scene, while the other
encoder, VDE, learns to encode the view-dependent aspects of the
scene. Taking the two encodings together, a decoder can reconstruct
the input scene as if it was observed from the viewpoint represented
in the other scene.

The VIE is then used to train a world model that can be applied for
policy learning or behavior cloning. The results indicate that the
approach is able to disentangle the different aspects of the view
inputs, leading to improved generalization to viewing changes between
training and testing.

**Strengths:**

- The training setup to enforce a separation of view-independent and view-dependent encoders is clever and seems novel.

- The view generation results indicate that the encoders and the decoder learn the kind of representation intended by the authors.

- The evaluation of the learned policies indicates strong improvements over prior approaches to world model learning with respect to
generalization to novel viewpoints.

**Weaknesses:**

The main weakness of the paper lies in the experimental evaluation. As I understand, the model needs to learn (at least) two key aspects for a scenario: Where is the camera relative to the manipulator base (VDE) and what is the task-relevant layout of the scene (VIR). The decoder then needs to use that information to generate a novel viewpoint of the input scene. While this seems to work for the test cases, it is not clear whether the success is due to overfitting to a rather small number of settings and tasks, or whether the approach would scale to more complex scenarios, including real world data and camera pose changes beyond azimuth. The current evaluation does not provide sufficient evidence regarding real world significance.

In light of changing camera poses, the action space of the controller is very important. There are at least three I could imagine: (delta) joint space, (delta) end effector pose in camera frame of reference, (delta) end effector pose in manipulator frame of reference. The specific choice is extremely important for how a policy might transfer to different camera viewpoints. Which specific one was used in the experiments? Could the approach work for all of these? What if the camera calibration of a test scene is known?

**Questions:**

Additional questions beyond those already mentioned in the weakness sections.

I like the idea of training the encoder / decoder using multiple viewpoints. Using a different scene along with a different camera pose to training view dependent aspects seems elegant. However, I'm not sure that this is really necessary. What is there to be encoded beyond the pose of the camera relative to the manipulator? If there's nothing else, is training the VDE overkill to extract that information from the scene? It'd be interesting to validate what specifically is learned by the VDE (e.g. could you train a readout network to extract relative camera pose?). Also, does this work equally well for more translation and rotation of the camera?

Why does the model performance drop so significantly in the BC results shown in E.2? I can imagine that this might be due to the reference framework used for the robot actions.

A discussion in the context of 3D representations for keyframe-based manipulation would be helpful. Several works, such as [1,2], train a model that is independent of the camera viewpoint by generating actions in the camera frame of reference and then translating these into robot actions using a calibrated camera pose. Could the same idea be applied here?

[1] Perceiver Actor: https://peract.github.io/
[2] 3D Diffusor Actor: https://3d-diffuser-actor.github.io/

---

> ### Author Response · Authors · 2024-11-22
> **Author Response (Part 1/2)**
>
> Thanks for your detailed comments and acknowledgement on the idea. Please find our response regarding the application scope and the experiments.
>
> > Q1: The main weakness of the paper lies in the experimental evaluation... Whether the approach would scale to more complex scenarios, including real world data and camera pose changes beyond azimuth.  The current evaluation does not provide sufficient evidence regarding real world significance.
>
> A1: Good point. We acknowledge the importance of demonstrating the robustness of our ReViWo framework in more challenging and realistic settings. To address this concern, we have conducted two additional experiments: (1) a real-world experiment using the ALOHA robot, and (2) an evaluation that includes variations in camera elevation.
>
> **For real world experiments**, we consider a long-term manipulation task that involves three stages: reaching for a bottle (stage 1), grasping it (stage 2), and then placing it on a plate (stage 3). We collected a dataset of 128 trajectories using three cameras (two thrid-person cameras and one gripper camera). The dataset includes 10 distinct types of bottles and plates, which were placed randomly within the operational area during data collection. We implement ReViWo-BC as follow: training VIE on the real world images and conducting behavior cloning on the control data with VIE representation. In this experiments, the policy inputs with all three camera images. To simulate viewpoint disturbances, we adjusted the azimuth by +15 degrees and the elevation by -15 degrees on one of the third-person cameras. We evaluate the method on both training and CIP viewpoint for 10 trajectories with random initalization. Experimental results are as follow:
>
> | Success Rate         | Stage 1 | Stage 2 | Stage 3 |
> | -------------------- | ------- | ------- | ------- |
> | ReViWo-BC (Training) | 100%    | 80%     | 60%     |
> | ReViWo-BC (CIP)      | 100%    | 60%     | 50%     |
>
> Refer to [this link](https://anonymous.4open.science/r/ReViWo-448B/assets/real_world.gif) for the real world deployment video. The results indicate that the proposed method maintains robust control when subjected to novel viewpoints in a real-world setting. This result serves as a preliminary evidance of the applicability of ReViWo method.
>
> **For evaluation under camera elevation variations**, we present the results as follow:
>
> | Eva. Offset | 0        | -2     | -4       | -6       | -8       | -10      | -12      |
> | ----------- | -------- | ------ | -------- | -------- | -------- | -------- | -------- |
> | ReViWo      | **98.9** | **90** | **97.2** | **72.8** | **51.1** | **40.6** | **18.3** |
> | COMBO       | 98.9     | 36.1   | 11.7     | 29.4     | 1.1      | 1.7      | 0        |
>
> The results present that ReViWo's generalization is not only limited to the azimuth change, showcasing its applibility.
>
> > Q2: In light of changing camera poses, the action space of the controller is very important...
>
> This is an important point in robotics. We demonstrate ReViWo can be applied to various settings. We answer your questions below.
>
> **Comment 1:** Which specific one was used in the experiments? Could the approach work for all of these?
>
> **Response 1:** In current evaluations, we use **both** joint space (real world ALOHA robot) and end effector in manipulator frame of reference (MetaWorld and PandaGym). The results present ReViWo work for all these two common settings (i.e., more performance under viewpoint disturbance compared to baselines). We suggest that ReViWo is applicable to these settings, because the key aspect of ReViWo is the decomposition of visual observations into view-invariant and view-dependent representations, which can be applied regardless of the specific action space.
>
> **Comment 2:** What if the camera calibration of a test scene is known?
>
> **Response 2:** We believe this could be a new setting and this paper focuses on visual control without camera calibration. With known camera calibration, it could potentially enhance the performance of ReViWo. For example, the transformation between the camera frame and the manipulator frame can be accurately determined and more techniques using camera calibration can be involved, allowing for more precise control and potentially improving the robustness of the policy under varying viewpoints. This could be an interesting direction for further improving the adaptability of our method.

---

> > ### Comment · Reviewer_YacQ · 2024-11-25
> > **Real world experiments**
> >
> > Thanks for the real world experiments. To be honest, I think that these are still pretty simple, and various other techniques have been shown to be able to handle pick and place with varying object positions. Why did you only change the position of one of the two cameras? I assume you tried varying both? In this context it'd also be interesting how well your technique works without gripper camera.

---

> ### Author Response · Authors · 2024-11-22
> **Author Response (2/2)**
>
> > A3: What is there to be encoded beyond the pose of the camera relative to the manipulator? If there's nothing else, is training the VDE overkill to extract that information from the scene? It'd be interesting to validate what specifically is learned by the VDE. Also, does this work equally well for more translation and rotation of the camera?
>
> A3: We clarify the training of VDE is to effective separate view-invariant information from the visual input (We now provide a mathematical justification for the representation separation in Appendix F in the revised version). The VDE encodes not only the camera pose relative to the manipulator but also captures other view-dependent features such as lighting, shadows, and background variations. For empirical evidence of the VDE's capability to encode complex background information that the VIE does not, please refer to [this link](https://anonymous.4open.science/r/ReViWo-448B/assets/real_world_reconstruct.png) for real-world examples. Besides, we also demonstrate that that the VDE effectively generalizes to novel inputs not encountered during training ([link](https://anonymous.4open.science/r/ReViWo-448B/assets/metaworld_novel.png)), showcasing its robustness to variations in background and camera rotation/translation. The VDE's ability to generalize well to new viewpoints suggests that its training is not overkill but essential for the representation separation.
>
> > Q4: Why does the model performance drop so significantly in the BC results shown in E.2? I can imagine that this might be due to the reference framework used for the robot actions.
>
> A4: This is a noticeable result. We suggest that the performance drop is primarily attributed to distributional shift and the accumulation of errors during sequential decision-making. BC is inherently susceptible to these issues, as it relies on imitating actions from a fixed dataset. However, it is noteworthy that when BC is paired with VIR, it outperforms BC with VAE representations. This outcome underscores the robustness of VIR in novel viewpoints and improving overall performance.
>
> > Q5: A discussion in the context of 3D representations for keyframe-based manipulation would be helpful... Could the same idea be applied here?
>
> A5: Thanks for your suggestions. 3D representations may provide some structural information, which improves the manipulation and generalization. We suggest that 3D setting could be a good supplement to this work. For your proposed work, PerACT [1] takes the voxel grid as input and 3D Diffusor [2] takes the tokenized 3D representation as input, require more than one multi-view RGBD images simultaneously and also the corresponding camera parameters. We would like to add a section discussing the works about 3D representations.
>
> #### Reference
>
> [1] Perceiver-Actor: A Multi-Task Transformer  for Robotic Manipulation. Shridhar et al. 2022.
>
> [2] 3D Diffuser Actor: Policy Diffusion with 3D Scene Representations. Ke et al. 2024.
>
> ------
>
> Thanks again for the careful response. We are **glad to any further discussions**.

---

> ### Author Response · Authors · 2024-11-25
> **Response to follow-up comments**
>
> Thanks for your timely response. For your follow up comment:
>
> > Q: Why did you only change the position of one of the two cameras? I assume you tried varying both?
>
> A: We clarify that **we did not conduct experiments on changing both cameras' viewpoints**. In previous experiments, we change only one viewpoint because another one camera is fixed on the robot body by screws.
>
> **Evaluation on more challenging setting**: For more comprehensive evaluation, we have managed to change the both third-person cameras' viewpoints using tape. The new CIP setting now becomes: {Camera 1: azimuth+15, elevation-15; Camera 2: azimuth+15, elevation-15}. To better present the robustness of our method, we introduce a new baseline: ACT [1] (Action Chunking Transformer), for comparison. ACT performs end-to-end imitation learning directly from real demonstrations, with pre-trained ResNet-50 as the vision encoder. The additional results are as follow:
>
> | Success Rate         | Stage 1 | Stage 2 | Stage 3 |
> | -------------------- | ------- | ------- | ------- |
> | ReViWo-BC (Training) | 100%    | 80%     | 60%   |
> | ReViWo-BC (New CIP)      | 100%    | 60%   | 60%  |
> | ACT (Training)      | 100%    | 60%   | 50%  |
> | ACT (New CIP)      | 100%    | 0%   | 0%  |
>
> We observe that ReViWo-BC maintains its performance robustness in the new, more challenging CIP setting. In contrast, ACT performs comparably to ReViWo-BC on training viewpoint, while exhibiting a significant decline in performance when viewpoint changes. These additional results further substantiate the efficacy of the our method in handling viewpoint disturbances.
>
> ------
>
> We hope these additional results could address your concern. We are glad for further discussions.
>
> ### Reference
>
> [1] Learning Fine-Grained Bimanual Manipulation with Low-Cost Hardware. Zhao, et al.

---

### Author Response · Authors · 2024-11-24
**General Response**

We express our gratitude to the reviewers and chairs for their valuable time and constructive feedback on this paper. We are encouraged to note that reviewers acknowledge this paper's novelty (Reviewer YacQ,juRH,Czra), its contribution to community (Reviewer Czra,Hd4G,pubk) and the sufficiency/performance of the experiments (Reviewer YacQ,juRH,Czra,Hd4G). Below we summarize the major concerns raised by reviewers and our corresponding response.

1. **Requiring real world evaluation** (Reviewer YacQ,juRH,Czra,Hd4G,pubk): The primary concern is the need for real-world evaluation. We have now included an evaluation using the real-world ALOHA robot, which confirms the successful application of ReViWo method in a real-world setting. The deployment video is at [link](https://anonymous.4open.science/r/ReViWo-448B/assets/real_world.gif).
2. **Evaluation with more diverse tasks and baselines** (Reviewer YacQ,Czra,Hd4G,pubk): We have conducted experiments on more diverse tasks (three tasks on Meta-World, including Coffee Button, Faucet Open and Dial Turn) in addition to three tasks in main body. Besides, we evaluate ReViWo with new baseline (RT-X), and on new type of viewpoint disturbance (i.e., elevation change).
[**UPDATE 29 Nov.**] We have extended the evaluations on 50 Meta-World tasks. Please refer to [our response](https://openreview.net/forum?id=vJwjWyt4Ed&noteId=oxChgwurah) to Reviewer Czra for details.
3. **Effectiveness of the representation separation** (Reviewer pubk) & **quality of the separation** (Reviewer YacQ,juRH,Czra): We have added a mathematical analysis to justify the effective disentanglement of the view-invariant and view-dependent representations. Additionally, we have provided more evidences of this separation through more decoder outputs on real-world images and novel visual inputs, accompanied by an analysis of the results.

We believe that the experiments are thorough and this paper makes significant contributions to research community, as **the viewpoint-robustness for robot is important and challenging** (suggested by Reviewer Czra and Hd4G). We have provided detailed responses to each reviewer's comments below. We ***are eager to receive further feedback*** and are ready to engage in discussion to address any additional questions or concerns.

------

Best wishes,

Submission #219 authors

---

> ### Author Response · Authors · 2024-11-26
> **Paper update and additional real world evaluations**
>
> Dear reviewers,
>
> Thanks for your time and valuable comment on this paper. We have made significant improvements to our paper based on the valuable feedback provided by the reviewers. We highlight the revised content in red in the [updated version](https://openreview.net/pdf?id=vJwjWyt4Ed) for your convenience. We believe that the paper is now more polished and easier to understand. Here are the main updates:
>
> ### Paper update
> 1. Additional experiments about real world robot have been conducted to resolve the concerns raised. Figure 3 and Figure 4, and Table 1 are updated to reflect these changes. (Reviewer YacQ,juRH,Czra,Hd4G,pubk)
> 2. Section 1 (Introduction) now integrates new empirical results and discussion on learning representations from 3D data. (Reviewer YacQ)
> 3. We add a section (Appendix F) to provide mathematical justification that the optimization objective can facilitate the separation of view-invariant and view-dependent representations. (Reviewer pubk)
> 4. The paper is re-organized, moving Section 5 (related work) to Appendix B.
> 5. Section 4 (Experiment) is updated to integrate more comprehensive discussions and accurate statements on the experimental results. (Reviewer juRH,Czra)
> 6. Section 3 (Method) and Section 4 (Experiment) now includes the links to the Appendix for readers to better obtain supplement information.
>
> ### Additional real world results
> We conduct real world experiments on more challenging setting, with a new real world baseline for comparison. Please refer to [the response to Reviewer YacQ](https://openreview.net/forum?id=vJwjWyt4Ed&noteId=IzixRs2wMz) and Section 4.2 in the updated paper for more details.
>
> ------
>
> Best wishes,
>
> Submission #219 authors

---

### Author Response · Authors · 2024-12-03
**Kindly request feedback**

Dear reviewers,

We appreciate the time and effort you have dedicated to reviewing this paper and offering detailed feedback. As the discussion period draws to a close, we kindly request confirmation on whether any issues or questions persist that require further clarification. We are open to and welcome additional discussions regarding this work.

Best regards,

Submission 219 authors.

---

### Meta-Review · Area_Chair_smrE · 2024-12-18

**Metareview:**

This paper presents ReViWo, a framework for learning view-invariant world models for visual robotic manipulation that aims to maintain performance under camera viewpoint changes. The paper received mixed reviews with scores ranging from 3 to 6, with most reviewers finding value in the core idea but raising concerns about evaluation scope and technical depth.

The reviewers consistently praised the paper's novel approach to handling viewpoint robustness through representation learning, its clear writing, and its promising initial results. In particular, reviewers YacQ, juRH, and Czra highlighted the elegance of separating view-invariant and view-dependent information, while Hd4G noted the practical importance of handling viewpoint changes in robotics.

Initial concerns centered on: limited real-world evaluation, insufficient task diversity in experiments, and lack of mathematical justification for the representation separation. The authors provided substantial responses during the rebuttal period, conducting new real-world experiments with an ALOHA robot on a multi-stage manipulation task, extending evaluations to 50 Meta-World tasks, and adding mathematical analysis of the representation learning objective.

These responses satisfied several reviewers, with YacQ and Hd4G explicitly increasing their scores. However, reviewer pubk, who gave the lowest score of 3, raised fundamental concerns about the approach's practicality compared to depth-based solutions and questioned the technical novelty beyond existing methods like COMBO. Despite detailed responses from the authors addressing these points, including clarifications about the limitations of depth-based approaches and new mathematical analysis justifying their method, pubk did not engage further in the discussion or update their assessment.

The remaining reviewers, while acknowledging some limitations, found the authors' responses satisfactory. Czra, though maintaining their score, suggested that more comprehensive evaluation would strengthen the work but viewed the research direction as promising.

The consensus view, considering the engaged reviewers' feedback, appears to be that while ReViWo may have limitations and would benefit from more extensive evaluation, it represents a valuable contribution to view-robust manipulation. The core technical idea is sound and shows promise in both simulation and real-world settings.

**Additional Comments On Reviewer Discussion:**

None -- see metareview

---

### Decision · Program_Chairs · 2025-01-22

Accept (Poster)